# Mesoscale Dynamics and Eddy Heat Transport in the Japan/East Sea from 1990 to 2010: A Model-Based Analysis

**Dmitry Stepanov** [1,*] , **Vladimir Fomin** [2,3] , **Anatoly Gusev** [2,3,4] and **Nikolay Diansky** [2,3,5]

1   Department of the Ocean and Atmosphere Physics, V.I.Il'ichev Pacific Oceanological Institute of FEB RAS, 690041 Vladivostok, Russia
2   N.N.Zubov's State Oceanographic Institute, Roshydromet, 119034 Moscow, Russia; vladimirfomin@live.com (V.F.); anatoly.v.gusev@gmail.com (A.G.); nikolay.diansky@gmail.com (N.D.)
3   Marchuk Institute of Numerical Mathematics of the Russian Academy of Sciences, 119333 Moscow, Russia
4   P.P. Shirshov Institute of Oceanology of the Russian Academy of Sciences, 117997 Moscow, Russia
5   Faculty of Physics, Lomonosov Moscow State University, 119991 Moscow, Russia
*   Correspondence: step-nov@poi.dvo.ru; Tel.: +7-924-230-8382

**Abstract:** The driving mechanisms of mesoscale processes and associated heat transport in the Japan/East Sea (JES) from 1990 to 2010 were examined using eddy-resolving ocean model simulations. The simulated circulation showed correctly reproduced JES major basin-scale currents and mesoscale dynamics features. We show that mesoscale eddies can deepen isotherms/isohalines up to several hundred meters and transport warm and low salinity waters along the western and eastern JES boundaries. The analysis of eddy kinetic energy (EKE) showed that the mesoscale dynamics reaches a maximum intensity in the upper 300 m layer. Throughout the year, the EKE maximum is observed in the southeastern JES, and a pronounced seasonal variability is observed in the southwestern and northwestern JES. The comparison of the EKE budget components confirmed that various mechanisms can be responsible for the generation of mesoscale dynamics during the year. From winter to spring, the baroclinic instability of basin-scale currents is the leading mechanism of the JES mesoscale dynamics' generation. In summer, the leading role in the generation of the mesoscale dynamics is played by the barotropic instability of basin-scale currents, which are responsible for the emergence of mesoscale eddies, and in autumn, the leading role is played by instabilities and the eddy wind work. We show that the meridional heat transport (MHT) is mainly polewards. Furthermore, we reveal two paths of eddy heat transport across the Subpolar Front: along the western and eastern (along 138° E) JES boundaries. Near the Tsugaru Strait, we describe the detected intensive westward eddy heat transport reaching its maximum in the first half of the year and decreasing to the minimum by summer.

**Keywords:** eddy-resolving numerical simulations; the Japan/East Sea; mesoscale eddies; meridional and zonal eddy heat transport; baroclinic and barotropic instability



## 1. Introduction

Heat uptake and redistribution in the ocean are the major subjects of physical oceanography [1,2]. There are two heat transport mechanisms. One is associated with large-scale currents, while the other is associated with mesoscale eddies [3–6]. Chains of eddies stir isotherms around [7], and an individual eddy can transport warm water inside itself far from its birth location [8]. Due to the numerous mesoscale eddies [9] in the ocean, their impact on heat transport and redistribution can be significant [10]. However, because of sporadic and short-term in situ measurements and satellite observations of the sea surface only, the estimation of the mesoscale eddy impact on the heat transport is still far from complete.

To reveal the mechanisms of the mesoscale dynamics' generation, the approach based on the analysis of the EKE budget components is generally used [11,12]. Studies based on

satellite altimetry have led to the conclusion that the central mechanism of the mesoscale dynamics' generation is associated with the instability of large-scale currents, in particular baroclinic instability [13]. This suggestion was entirely confirmed by studies of the mesoscale dynamics' generation based on high-resolution numerical modeling and EKE budget analysis in various regions of the World Ocean [11,12,14,15]. At the same time, there are World Ocean regions where the barotropic instability of large-scale currents plays a leading role in the mesoscale dynamics' generation [16,17]. The analysis of the EKE budget components found that for marginal seas, excluding the instability of the basin-scale currents, a leading role in the mesoscale dynamics' generation can be played by other EKE sources, e.g., the wind work [18]. This component, together with the baroclinic instability, is responsible for the winter intensification of mesoscale dynamics on the Eastern Sakhalin shelf in the Sea of Okhotsk [19]. Due to expensive downscaling numerical simulations, the spatial resolution of the global ocean models is eddy-permitting at high latitudes and in marginal seas. Thus, we need regional eddy-resolving numerical simulations, in particular at high latitudes, where there are obstacles for the global ocean modeling and in situ observations of mesoscale dynamics induced by the significantly decreasing spatial size of mesoscale eddies in contrast to the mid- and low latitudes [20,21].

The Japan/East Sea (JES) is a marginal sea at high latitudes in the northwestern Pacific Ocean [22]. The JES circulation is driven mainly by the water exchange through the shallow passages and the Asian monsoons [22]. The JES circulation features and intense mesoscale variability are studied with in situ and satellite observations, drifter's trajectories observations, as well as with numerical modeling. Isoda [23] analyzed thermal maps at a depth of 200 m in the southeastern part of the JES from 1958 to 1992 and established that warm mesoscale eddies that were regularly generated here then were propagated eastward. Gordon et al. [24] revealed the generation of intrathermocline eddies in the southeastern JES. These eddies deepened isotherms up to 100 m, and due to the compensatory baroclinicity of the upper and lower layers, the sea level anomalies were small. Lee and Niiler [25], based on the AVISO altimetry observations, found strong eddy dynamics, where the East Korea Warm Current (EKWC) was. The spatial size of such mesoscale eddies varies from 38 km to 60 km. Based on observations of drifter trajectories, Lee et al. [25] estimated the surface EKE and found that it reached its maximum in the southern JES, while its minimum was observed in the northern JES. In addition, they revealed that EKE reached its maximum along the JES northwestern boundary, where the cold Primorye Current was. Prants et al. [26] carried out a comprehensive study of mesoscale dynamics in the Primorye Current based on Lagrangian trajectory analysis. They revealed the long eddy chain generated in the Primorye Current. Ostrovskii et al. [27] investigated turbulent mixing on the continental slope where the Primorye Current was. They found that high vertical turbulent diffusivity was associated with mesoscale eddies. Based on the AVISO altimetry-based velocity, Prants et al. [28] found cases of subtropical water inflow from the southern to the northern JES across the Subpolar Front (SF). Lee et al. [29] presented a comprehensive study of the mesoscale eddy's spatial and temporal scales in the JES. Specifically, they indicated that mesoscale eddies' lifetime could vary from 1–3 mo, and some eddies could live up to 6 mo. Satellite and in situ observations, as well as drifter trajectories allowed us to reveal a complicated pattern of the JES mesoscale dynamics. Together with these studies, the investigations were carried out based on high-resolution regional numerical modeling.

Numerical simulation was successfully used to reproduce the JES mesoscale dynamics. Hogan and Hurlburt [30] carried out sensitivity studies and established that directly taking into account the mesoscale dynamics in numerical simulations was necessary to correctly reproduce the EKWC separation and intensive circulation in the intermediate and abyssal layers. Jacobs et al. [31], based on altimetry observations and eddy-resolving numerical simulations, estimated the Reynolds stress induced by the mesoscale turbulence. They established that mesoscale variability was promoted in the EKWC separation from the southwestern JES boundary and the transport of the SF. However, the above-mentioned

studies did not reveal the leading mechanisms of the mesoscale dynamics' generation and heat transport induced by mesoscale eddies.

The studies of heat uptake and redistribution in the JES were carried out based on in situ observations [32,33]. The authors of these studies found out that the primary cause of heat content variations in the JES upper layer (from the surface to 300 m) was associated with water exchange through the Korea/Tsushima Strait. However, the authors used temperature data mainly from the southern JES and did not estimate the impact of mesoscale dynamics on heat content variations.

In the present study, we investigated the JES mesoscale dynamics based on eddy-resolving numerical simulation. We considered the mechanisms of its generation based on the comparative analysis of the EKE budget components and estimated the horizontal heat transport induced by the mesoscale dynamics and its seasonal variations.

Section 2 presents a description of the numerical model configuration and the setup of the experiments. The validation of the simulation results based on the comparison with satellite altimetry AVISO, GOFS3.1 reanalysis, and in situ observations is presented in Section 3. The features of the simulated mesoscale dynamics, the analysis of the EKE budget components, and the heat transport induced by the JES mesoscale dynamics are presented in Section 4. The discussion and conclusions are presented in Sections 5 and 6, respectively.

## 2. Model Configuration and Setup of the Simulations

Based on retrospective numerical simulations, we investigated the heat transport induced by the mesoscale dynamics in the JES. These numerical simulations were carried out with the ocean general circulation model (OGCM) INMOM (Institute of Numerical Mathematics Ocean Model). The INMOM is based on the ocean hydrothermodynamics equation set with hydrostatic and Boussinesq approximations and a terrain-following vertical coordinate $\sigma$. The basic equations and numerical implementation were presented in [34]. The INMOM was successfully used to simulate the circulation of the World Ocean [35,36] and marginal seas [19,37,38]. The authors of [39,40] carried out retrospective numerical simulations of the JES circulation based on the INMOM taking into account heat and salt exchange through the JES Straits only. Since the JES circulation is derived by the water exchange through the JES Straits [22], open boundary conditions must be formulated in these simulations. In [41,42], the modified INMOM configuration was used, where this water exchange was naturally formed.

In this study, we used the modified INMOM configuration with a higher spatial resolution and atmospheric parameters from the ERA-Interim dataset. The model domain covers not only the JES, but also the Yellow Sea, East China Sea, the southwestern part of the Sea of Okhotsk, and the northwestern part of the Pacific Ocean (Figure 1). Such a model domain is extensive enough to take into account the influence of the Kuroshio, as well as the Yellow and East China Seas, on the water exchange through the Korea/Tsushima Strait. Furthermore, we took into account the sea level difference between the JES and the Sea of Okhotsk and reproduced the exchange through the Soya Strait [43].

The model's horizontal resolution was $1/20°$ in latitude and longitude. The estimation of the first baroclinic Rossby deformation radius ($\lambda_1$) [21,44] yielded that $\lambda_1$ varies from 14 km to 18 km in the southern JES and from 5 km to 10 km in the northern one (not shown). Since the typical spatial size of mesoscale eddies varies from 2–3 $\lambda_1$ [29], the used model's spatial resolution is enough to resolve the mesoscale dynamics of almost the whole JES explicitly. The vertical $\sigma$-levels were refined near the surface and bottom to reproduce the boundary layers. The topography used in the INMOM was obtained from the ETOPO1 dataset [45]. It was interpolated on the model grid and smoothed by a nine-point smoothing three times.

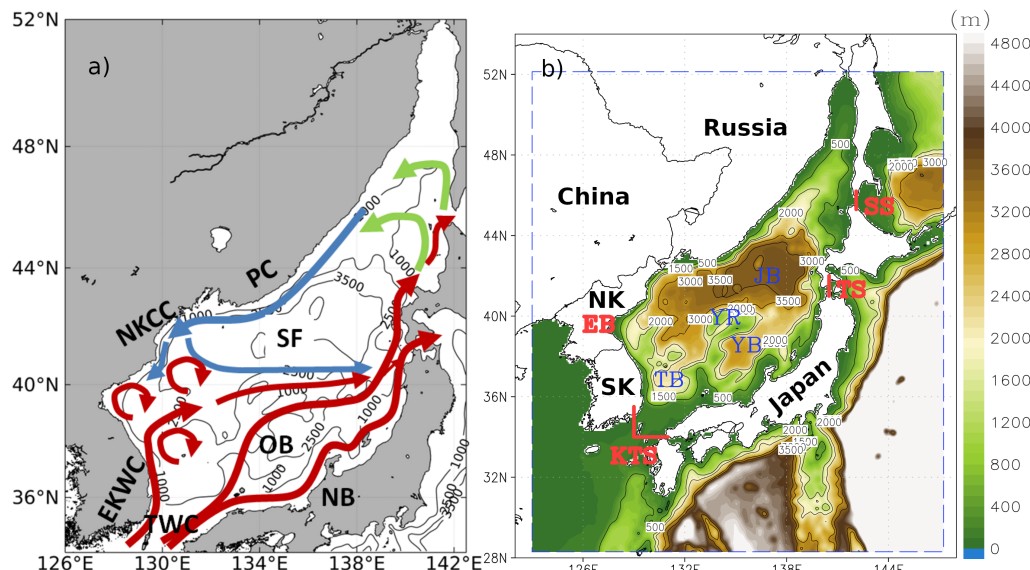

**Figure 1.** (**a**) The schematic pattern of the surface circulation from drifter observations [25]. Red arrows denote warm currents and eddies; blue arrows denote cold currents; green arrows denote neutral currents. TWC denotes the Tsushima Warm Current; EKWC denotes the East Korea Warm Current; OB denotes the Offshore Branch of the TWC; NB denotes the Nearshore Branch of the TWC; SF denotes the Subpolar Front; PC denotes the Primorye Current; NKCC denotes the North-Korea Cold Current. (**b**) Bathymetric map of the Japan/East Sea on the model grid (1/20°) based on the ETOPO1 dataset [45]. The dashed blue line denotes the model domain. Red segments denote sections where the transport through the straits was estimated. JB denotes the Japan Basin; YB denotes the Yamato Basin; TB denotes the Tsushima Basin; YR denotes the Yamato Rise; KTS denotes the Korea/Tsushima Strait; TS denotes the Tsugaru Strait; SS denotes the Soya Strait; EB denotes the Eastern Bay; NK denotes North Korea; SK denotes South Korea.

To parameterize the sub-grid processes, we used the Laplace-like operators, with the vertical mixing parameterized by the Pacanowski–Philander technique [46], where the vertical diffusivity and viscosity depend on the Richardson number. The vertical convection was parameterized by setting sufficiently high values for the vertical diffusivity/viscosity. The main parameters of the model configuration are presented in Table 1.

**Table 1.** Overview of the INMOM configuration.

| | |
|---|---|
| Model domain | 123° E–147.25° E, 28.3° N–52.12° N |
| Topography | ETOPO1 |
| Horizontal resolution | $1/20° \times 1/20°$ |
| Vertical resolution | 25 $\sigma$-levels |
| Hindcast period | 1979–2011 |
| Mixing technique | Laplace-like operator |
| Heat and salt lateral diffusivity | $5 \, \mathrm{m^2 \, s^{-1}}$ |
| Lateral harmonic viscosity | $10 \, \mathrm{m^2 \, s^{-1}}$ |
| Vertical mixing parameterization | Pacanowski–Philander method |
| Vertical diffusivity of salt and heat | $10^{-6} \, \mathrm{m^2 \, s^{-1}}$ |
| Vertical viscosity | $10^{-4} \, \mathrm{m^2 \, s^{-1}}$ |
| Convective mixing is parameterized by enhanced vertical diffusivity | $0.005 \, \mathrm{m^2 \, s^{-1}}$ |
| and vertical viscosity | $0.025 \, \mathrm{m^2 \, s^{-1}}$ |

Atmospheric forcing consists of heat, freshwater, and momentum fluxes. Since the IN-MOM version used has only the linearized free-surface condition without direct freshwater exchange, the artificial salt flux was used to take into account the freshwater effect. These fluxes were estimated by the following technique [47]. Note that the wind stress takes into account the sea surface velocity [48] to properly reproduce the mesoscale dynamics. The atmospheric parameters were used from the ERA-Interim reanalysis [49], which covers the period from 1979 to 2011. The 2 m air temperature and specific humidity, 10 m wind speed, and sea level pressure have a time discrepancy of 6 h. The downwelling long- and short-wave radiation, precipitation, and river runoff are provided averaged over 12 h.

One of the JES circulation's features is winter sea ice formation in its northern part. Freezing and melting of the sea ice change the salt balance and modify the momentum flux on the sea surface. Therefore, we took into account the sea ice thermodynamics and dynamics based on [50].

As initial conditions for potential temperature and salinity, we used their January climatology taken from the WOA2013v2.0 dataset [51,52]. For velocity, we set the initial condition as the no-motion state. Due to uncertainties in atmospheric forcing, as well as the imperfection of the parameterizations used, the simulated temperature and salinity can be significantly biased from their climatology. To reduce these biases, we nudged the simulated sea surface temperature and salinity to their monthly climatology. Therefore, we added some relaxation terms to the heat and salt surface fluxes. Relaxation time scales for potential temperature and salinity are 1 mo and 3 mo per 50 m, respectively.

At solid boundaries, we applied no-flux conditions for temperature and salinity, no normal flow for velocity with free-slip at the lateral boundary, and quadratic bottom friction. Near open boundaries, we reserved regions with a width of one degree, where the simulated temperature and salinity were nudged to their monthly climatology from surface to bottom [36,41].

We simulated the JES circulation from 1979 to 2011 with a 10 y spin-up to achieve a quasi-established annual cycle. After this, we considered the simulated dataset from 1990 to 2010.

## 3. Model Validation

Before analyzing the JES mesoscale dynamics, we performed the model validation and compared the simulation outputs with in situ and satellite observations and a high-resolution ocean reanalysis (GOFS3.1):

- Monthly mean satellite altimetry from the AVISO dataset (https://www.aviso.altimetry.fr/en/data/data-access.html,accessedon1December2021) with a spatial resolution of 0.25° in longitude and latitude covering the time period from 1994 to 2010;
- Monthly mean sea surface height from the GOFS3.1 reanalysis [53] with a spatial resolution of 1/12° in longitude and latitude covering the time period from 1994 to 2010;
- Monthly climatological temperature from the East Asian Seas Regional Climatology (EAS dataset) [54] with a spatial resolution of 1/10° in longitude and latitude covering the time period from 1804 to 2013;
- Monthly mean transport estimations through the Korea/Tsushima Strait based on in situ observations from 1997 to 2007 [55];
- Annual mean transport estimations through the Tsugaru Strait [56] and the Soya Strait based on in situ observations [43,57].

### 3.1. Simulated Long-Term Mean Geostrophic Circulation in the Japan/East Sea

First, we compared the simulated long-term (1990–2010) mean geostrophic currents with those evaluated from the AVISO satellite altimetry and GOFS3.1 reanalysis. Since the JES circulation is primarily derived by the monsoon atmosphere and seasonal variations of water exchange across the JES straits [22], we considered it in February and July, since the

volume transport across the Korea/Tsushima Strait shows large/small values in winter–spring/summer–autumn ([22], pp. 33–34).

The long-term monthly mean geostrophic currents can be estimated as:

$$\langle u_g \rangle = -g/f(\partial\langle\eta\rangle/\partial y), \ \langle v_g \rangle = g/f(\partial\langle\eta\rangle/\partial x), \tag{1}$$

where $\eta$ is the sea surface height (SSH), $u_g$ and $v_g$ are the zonal and meridional components of geostrophic velocities, respectively, $g$ is the gravity acceleration, $f$ is the Coriolis parameter, and $\langle...\rangle$ denotes long-term monthly mean averaging. To quantitatively estimate the geostrophic circulation's intensity, we used the magnitude of velocity.

Figure 2 shows long-term mean geostrophic currents (1) for the three datasets in February. In the southern JES (southward of the SF (see Figure 1a)), the three branches of the Tsushima Warm Current (TWC) (Figure 1a) are reproduced in the simulated geostrophic circulation (Figure 2a). The Nearshore Branch of the TWC (NB) follows along the eastern boundary of the JES; the Offshore Branch of the TWC (OB) follows the Nearshore one towards the sea; the East Korea Warm Current (EKWC) follows along the western boundary of the JES and turns to the west at 38–39° N. The three branches of the TWC reproduced in the simulations were similar to those reproduced in the geostrophic circulation derived from the GOFS3.1 reanalysis (Figure 2b) and the AVISO dataset (Figure 2c). In addition, the intensity of the simulated geostrophic circulation is similar to those derived from other datasets. As for the differences, the anti-cyclonic eddy was reproduced in the simulated geostrophic circulation and AVISO one in the East Bay; however, this eddy was not reproduced in the geostrophic circulation from the GOFS3.1 dataset.

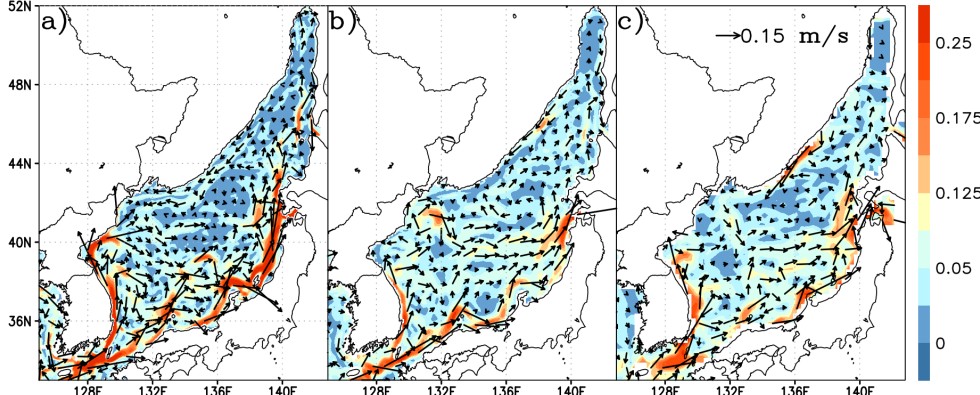

**Figure 2.** Long-term (1990–2010) mean geostrophic circulation ($\langle u_g \rangle, \langle v_g \rangle$) in the Japan/East Sea in February from INMOM simulations (**a**), GOFS3.1 reanalysis (**b**), and the AVISO dataset (**c**). Color denotes the magnitude of velocity (m s$^{-1}$).

In the northern JES (northward of the SF (see Figure 1a)), all the datasets produce the basin-scale cyclonic gyre in the long-term mean geostrophic circulation. This gyre manifests in the geostrophic circulation due to the strong wind impact in winter [58,59]. One of the gyre's components is the Primorye Current, which consists of the nearshore and offshore branches. Due to the coarse spatial resolution of the geostrophic circulations based on the AVISO and GOFS3.1 datasets, the Primorye Current consists of just one branch. In simulated geostrophic currents, the SF was more northward than in geostrophic circulations from both datasets. In addition, the simulated velocity along the SF was less intensive than the estimated one from the AVISO and GOFS3.1 datasets. An analysis of the scatter plots and general cell-by-cell statistics, where $R$ is the correlation coefficient and $RMSE$ is the root-mean-squared error, showed that the $R$ value between the simulated magnitude velocity and one from the AVISO/GOFS3.1 dataset equaled 0.46/0.5, and the $RMSE$ was equal to 0.04 m s$^{-1}$ for both datasets.

Figure 3 shows long-term mean geostrophic currents (1) in July for all datasets. We observed the intensification of currents over the whole basin for all datasets due to the

increasing transport through the Korea/Tsushima Strait. The simulated geostrophic circulation was similar to those from the AVISO and GOFS3.1 dataset. In addition, an anti-cyclonic eddy (Wonsan Eddy) was successfully reproduced in the East Korea Bay as indicated by the simulated geostrophic circulation (Figure 3a) and the AVISO dataset (Figure 3c) [25,29]. However, this anti-cyclonic eddy was not reproduced in the geostrophic circulation from the GOFS3.1 reanalysis (Figure 3b). Quantitative comparisons of the simulated velocity magnitude showed that the *R* value between this magnitude and the one from the AVISO/GOFS3.1 datasets equaled 0.56/0.6, and the *RMSE* was equal to 0.04 m s$^{-1}$ for both datasets.

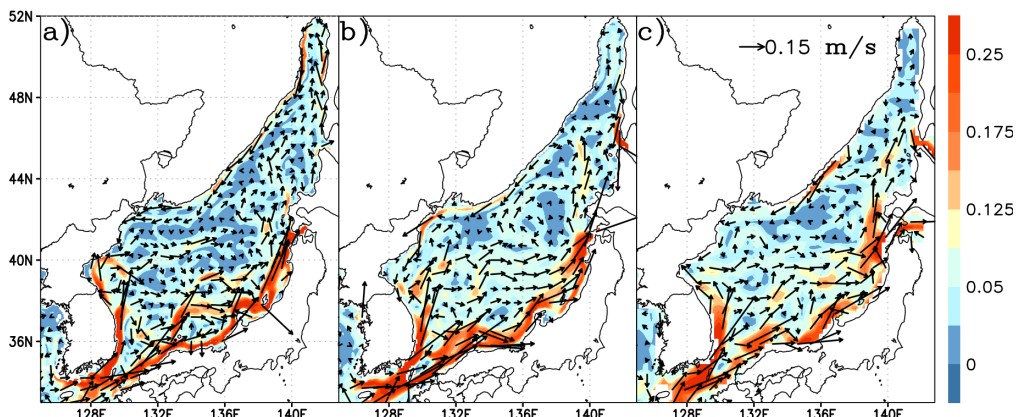

**Figure 3.** Long-term (1990–2010) mean geostrophic circulation ($\langle u_g \rangle, \langle v_g \rangle$) in the Japan/East Sea in July from INMOM simulations (**a**), GOFS3.1 reanalysis (**b**), and the AVISO dataset (**c**). Color denotes the magnitude of velocity (m s$^{-1}$).

### 3.2. Simulated Throughflow in the Japan/East Sea

Consider the throughflow in the JES, which is one of the primary reasons driving the JES circulation. We assessed the transport through the JES Straits (Table 2): the Korea/Tsushima Strait, the Tsugaru Strait, and the Soya Strait using the monthly mean simulated velocities on vertical transects across the straits (Figure 1b).

**Table 2.** Long-term mean throughflow in the JES assessed with the INMOM simulations, observations, and the GOFS3.1 reanalysis.

| Dataset (Period) | Korea/Tsushima Strait, Sv | Tsugaru Strait, Sv | Soya Strait, Sv |
|---|---|---|---|
| INMOM-ERA Int. (1990–2010) | 2.58 | 2.04 | 0.17 |
| Fukudome et al. [55](1997–2007) | 2.65 | – | – |
| Ito et al. [56] (1999–2000) | – | 1.51 | – |
| Ohshima et al. [43] (2003–2015) | – | – | 0.91 |
| Saveliev et al. [57](1975–1988) | – | – | 0.61 |
| GOFS3.1 (1994–2010) | 2.58 | 1.76 | 0.63 |

The long-term annual mean Korea/Tsushima Strait transport was similar to that estimated with the GOFS3.1 reanalyses. At the same time, the Korea/Tsushima Strait transport assessed from the simulated velocity was lower than that estimated from in situ observations [55]. This difference reached up to 0.07 Sv. Seasonal variations of the Korea/Tsushima Strait transport estimated with the simulated circulation showed that it reached its maximum (up to 2.86 Sv) from October to November, and its minimum (about

2.2 Sv) was observed from January to February. The Korea/Tsushima Strait transport estimated with the GOFS3.1 reanalysis reached its maximum (up to 3 Sv) in October, and its minimum (1.9 Sv) was observed in January. The difference between the maximum and minimum of this transport estimated with the simulated circulation was lower than the estimated one with the GOFS3.1 reanalysis. However, periods of the transport extrema for both datasets coincided with each other.

Long-term annual mean transport through the Tsugaru Strait exceeded 0.3 Sv assessed from the GOFS3.1 reanalysis and 0.5 Sv estimated from in situ observations [56]. Seasonal variations of this transport estimated with the simulated circulation showed that it reached its maximum (up to 2.3 Sv) from October to November, and its minimum (about 1.7 Sv) was observed from February to March. Ito et al. [56] estimated these variations as 1.1 Sv from October to November and 2.1 Sv from February to March. According to the GOFS3.1 reanalysis estimations, this transport reaches its maximum (up to 2 Sv) in November, and its minimum (about 1.7 Sv) is observed from January to February. These estimations of the transport through the Tsugaru Strait were close to those that estimated from the numerical simulations. The periods of the transport extrema for both datasets, as well as observations coincided with each other.

However, we observed the underestimation of the annual mean transport across the Soya Strait assessed from the simulated velocity field up to 0.46 Sv in comparison with the one estimated from in situ observations [57] and the GOFS3.1 reanalysis. The simulated transport reached its maximum (up to 0.32 Sv) from May to September, and its minimum (–0.1 Sv) was observed in January. Saveliev et al. [57] estimated seasonal variations of this transport from –0.01 to 1.18 Sv, and these variations estimated with the GOFS3.1 reanalysis varied from 0.1 Sv in January to 1.0 in August. We suggest that the underestimation of the annual mean transport across the Soya Strait resulted from the overestimation of the transport across the Tsugaru Strait. Nevertheless, the long-term mean difference between the inflow and outflow water from the numerical simulations reached 0.4 Sv, which is very close to that assessed from in situ observations (from 0.2 Sv to 0.53 Sv), and exceeded that estimated from the GOFS3.1 reanalysis (0.19 Sv).

### 3.3. Simulation-Based Potential Temperature

In this subsection, we analyze the long-term seasonal mean simulated potential temperature ($\theta$) on the sea surface and the depth of 100 m. In addition, we considered its vertical structure on the zonal and meridional sections. We compared the simulated $\theta$ with its climatology from the EAS dataset. We reduced the monthly mean simulated $\theta$ to match the one of the EAS dataset. For a simple visual comparison, we complement the scatter plots and general cell-by-cell statistics.

Figure 4 shows the long-term mean simulated $\theta$ and its climatology from the EAS dataset on the sea surface. In winter, the simulated $\theta$ was somewhat higher (1–2 °C) than that from the EAS dataset in the northwestern JES. However, in the southern JES, the simulated $\theta$ was somewhat lower (2–3 °C) than that from the EAS dataset. The SF from the simulated $\theta$ was shifted southward compared to that from the EAS dataset. Low *RMSE* and high *R* (see Figure 4c) reflect high similarity both maps.

In summer, we observed a higher conformity between both maps than in winter. *R* increased and *RMSE* decreased (see Figure 4f) compared to those in winter. In the southern JES, there is a higher similarity between both maps in contrast to the northwestern JES, where the simulated $\theta$ is higher than that from the EAS dataset (see Figure 4d).

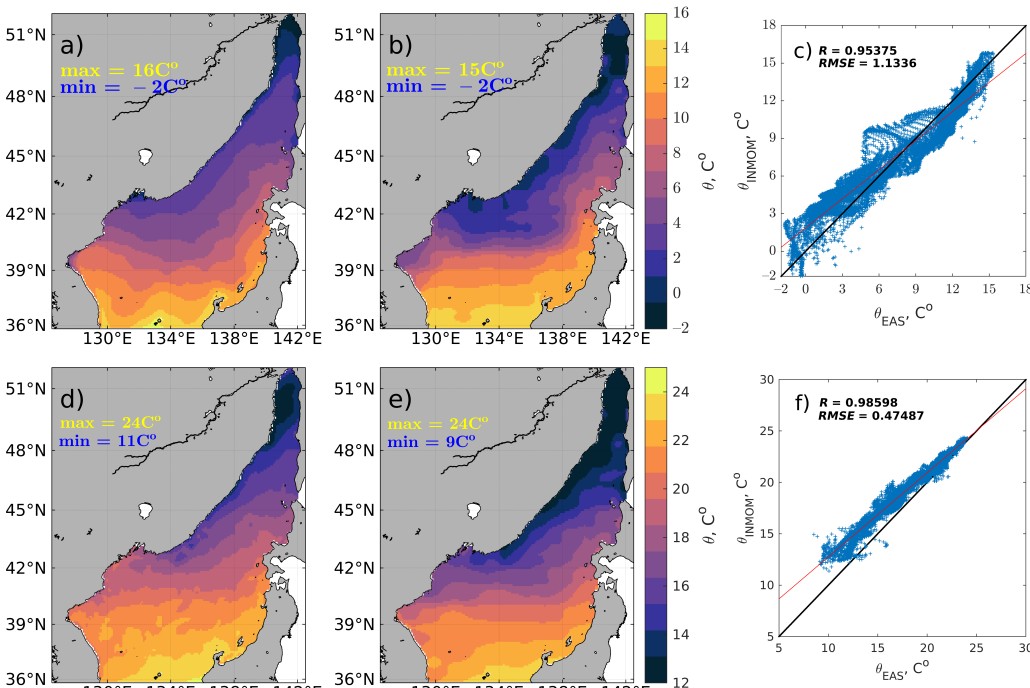

**Figure 4.** Long-term (1990–2010) mean sea surface potential temperature ($\theta_{\text{INMOM}}$) in the Japan/East Sea from: INMOM simulations (**a,d**), EAS dataset ($\theta_{\text{EAS}}$) (**b,e**); scatter plot with basic statistics: correlation coefficient (*R*) and root-mean-squared error (*RMSE*) (**c,f**) in winter (upper panels) and summer (lower panels).

At a depth of 100 m, there was a close correspondence between $\theta$ maps (see Figure 5). It was confirmed by the high value of *R* both in winter and summer. At the same time, the *RMSE* was higher than that on the sea surface. The simulated $\theta$ was higher than that from the EAS dataset in the northwestern JES.

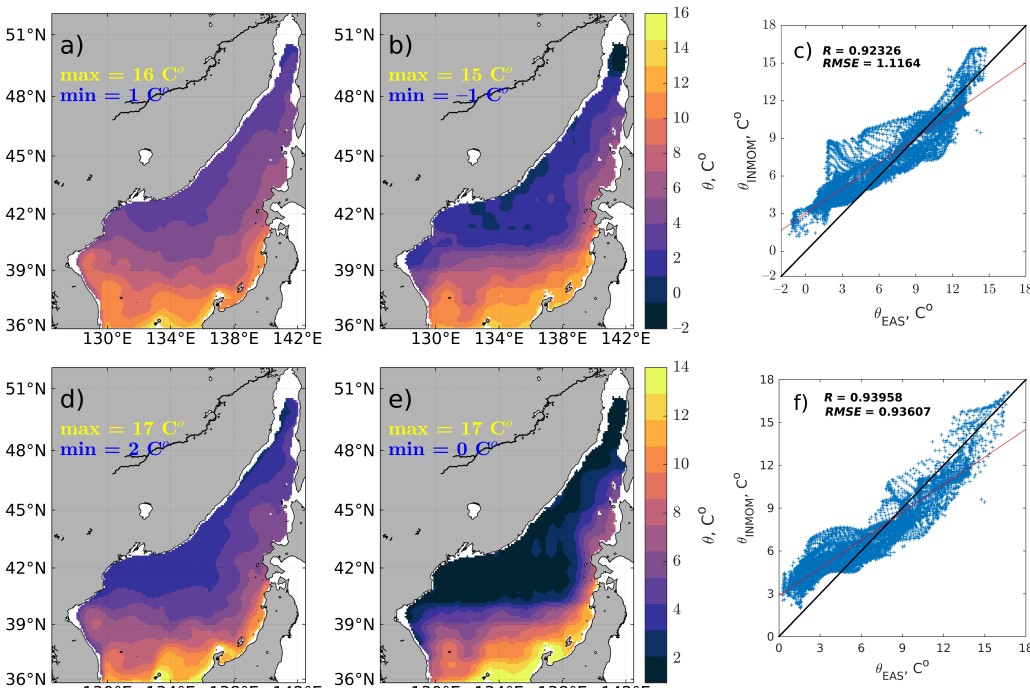

**Figure 5.** Long-term (1990–2010) mean potential temperature ($\theta_{INMOM}$) at a depth of 100 m from: INMOM simulations (**a**,**d**), EAS dataset ($\theta_{EAS}$) (**b**,**e**); scatter plot with basic statistics: correlation coefficient (*R*) and root-mean-squared error (*RMSE*) (**c**,**f**) in winter (upper panels) and summer (lower panels).

Let us consider the vertical structure of the simulated $\theta$ on the zonal and meridional sections, which intersect regions of major basin-scale currents of the JES, and compare with those from the EAS dataset. Figure 6 shows the vertical structure of the simulated $\theta$ and that from the EAS dataset on the zonal section along 37° N, which crosses three branches of the TWC. There is a closer correspondence between $\theta$ vertical structures both in winter and summer, which was confirmed by the high *R* value and low *RMSE* value (see Figure 6c,f). We observed three cores of the TWC branches both in winter and summer, which were characterized by higher $\theta$ and partial outcropping of isotherms.

On the meridional section along 134° E (see Figure 7), which crosses two branches of the TWC and the Primorye Current, the vertical structure of the simulated $\theta$ closely conformed to that from the EAS dataset, which was confirmed by the high *R* values and low *RMSE* values both in winter and summer (see Figure 7c,f). Note that the Primorye Current distinctly reflects both $\theta$ vertical sections in summer. There is an outcropping of isotherms in the northwestern JES. As for the discrepancies, we observed some higher $\theta$ values (2–3 °C) from the numerical simulations than those from the EAS dataset.

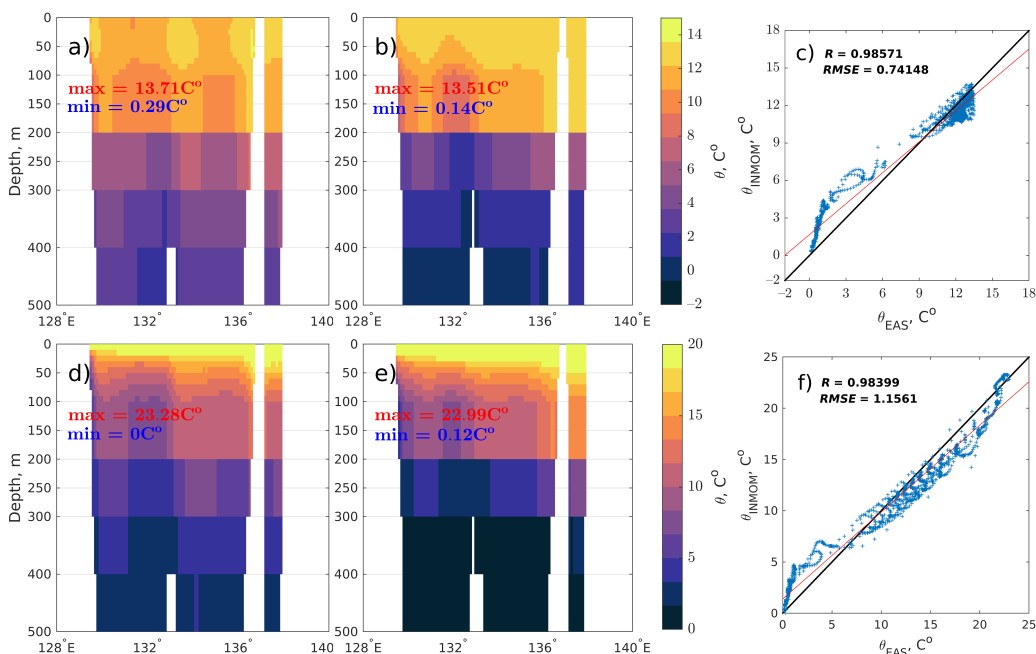

**Figure 6.** Long-term (1990–2010) mean potential temperature ($\theta_{\text{INMOM}}$) for the zonal section along 37° N from: INMOM simulations (**a,d**), EAS dataset ($\theta_{\text{EAS}}$) (**b,e**); scatter plot with basic statistics: correlation coefficient (*R*) and root-mean-squared error (*RMSE*) (**c,f**) in winter (upper panels) and summer (lower panels).

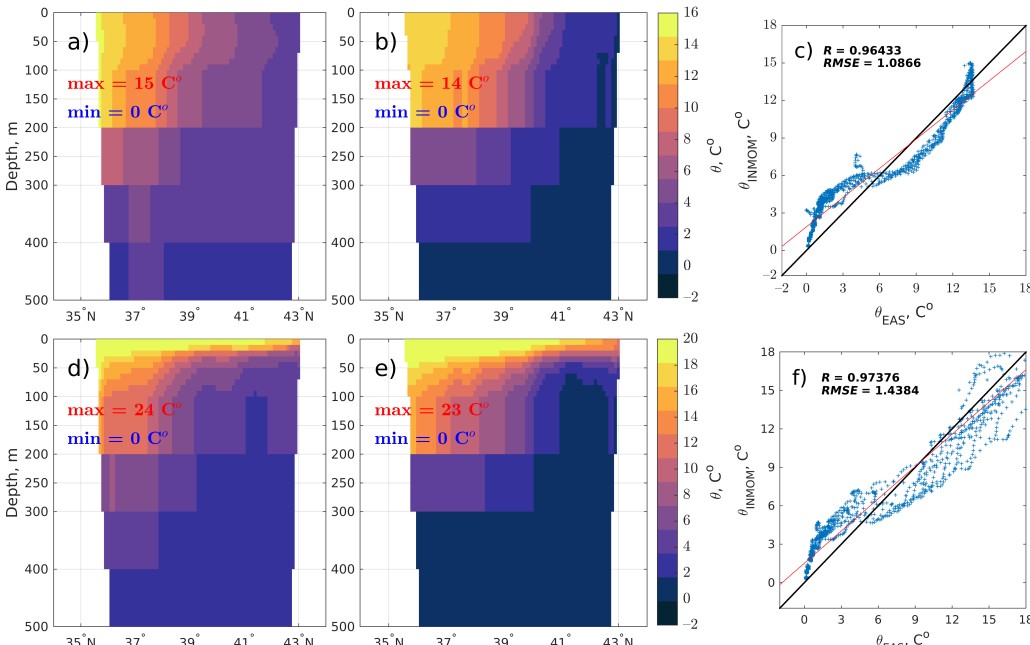

**Figure 7.** Long-term (1990–2010) mean potential temperature ($\theta_{\text{INMOM}}$) for the meridional section along 134° E from: INMOM simulations (**a,d**), EAS dataset ($\theta_{\text{EAS}}$) (**b,e**); scatter plot with basic statistics: correlation coefficient (*R*) and root-mean-squared error (*RMSE*) (**c,f**) in winter (upper panels) and summer (lower panels).

Thus, the long-term seasonal mean simulated $\theta$ conformed to the climatological values on the sea surface in winter. In summer, there was a closer correspondence between them. In depth, this correspondence remained. The vertical structures of the simulated $\theta$ on the zonal and meridional sections conformed to those from the EAS dataset both in winter and summer.

### 3.4. Simulation-Based Eddy Kinetic Energy

Since the major subject of this study is related to the mesoscale dynamics of the JES, we needed to evaluate the similarity of the simulated mesoscale dynamics with the estimate from the satellite datasets and reanalysis. For a qualitative assessment, we estimated the eddy kinetic energy per mass ($EKE_g$) calculated with geostrophic relations [13,60] as follows:

$$EKE_g = \frac{1}{2}\left(\overline{u_g'^2} + \overline{v_g'^2}\right), \tag{2}$$

where the overbar denotes a monthly mean averaging, $u_g' = -g/f(\partial\eta'/\partial y)$, $v_g' = g/f(\partial\eta'/\partial x)$, and the prime denotes deviation from the mean value.

To comprehensively investigate the mesoscale dynamics in the JES, we used the EKE estimations as follows [11,14,19]:

$$EKE = \frac{1}{2}\rho_0\left(\overline{u'^2} + \overline{v'^2}\right), \tag{3}$$

where $\mathbf{u}_h = (u, v)$ is the horizontal velocity field and $\rho_0$ represents the background density of seawater.

In order to confirm that $EKE_g$ was simulated correctly, we compared $EKE_g$ with that assessed from the AVISO dataset and GOFS3.1 reanalysis. Because the spatial resolution of the AVISO dataset has 0.25°, we reduced the simulated SSH and that from the GOFS3.1 reanalysis to match the AVISO dataset. We used the simple visual comparison in combination with the scatter plots and general cell-by-cell statistics.

Figure 8 shows the long-term annual mean $EKE_g$ ($aEKE_g$) assessed with the simulated SSH, AVISO dataset, and GOFS3.1 reanalysis. For all the datasets, $aEKE_g$ reached its highest values (more than $100\,cm^2\,s^{-2}$) in the southern JES, and its maxima were observed in the southeastern and southwestern JES. The $EKE_g$s assessed from the numerical simulations and AVISO dataset reached $193\,cm^2\,s^{-2}$ and $240\,cm^2\,s^{-2}$, respectively (Figure 8a,c), whereas $EKE_g$ estimated from the GOFS3.1 reanalysis (Figure 8b) did not exceed $131\,cm^2\,s^{-2}$ in the southern JES. At the same time, both $aEKE_g$ assessed from the simulated SSH and GOFS3.1 reanalysis showed high values compared to the $aEKE_g$ assessed from the AVISO dataset in the northwestern JES. These high $EKE_g$ values ($\approx 20\,cm^2\,s^{-2}$) are associated with the Primorye Current. This was confirmed by mesoscale dynamics studies in the northwestern JES [28]. In contrast, the $aEKE_g$ assessed from the AVISO dataset showed its minimum in the northwestern JES ($\approx 2\,cm^2\,s^{-2}$). These discrepancies resulted from the coarse spatial resolution of the AVISO dataset in the northwestern JES. Scatter plots show that the $aEKE_g$ assessed with simulated SSH closely conformed to both datasets, which was confirmed by the high $R$ value. At the same time, the $aEKE_g$ assessed with the simulated SSH showed lower values than the $aEKE_g$ assessed with the AVSIO dataset and higher values than the $aEKE_g$ assessed with the GOFS3.1 reanalysis. However, the $aEKE_g$ assessed with the simulated SSH closely conformed to that assessed from the GOFS3.1 reanalysis in the region of its small values.

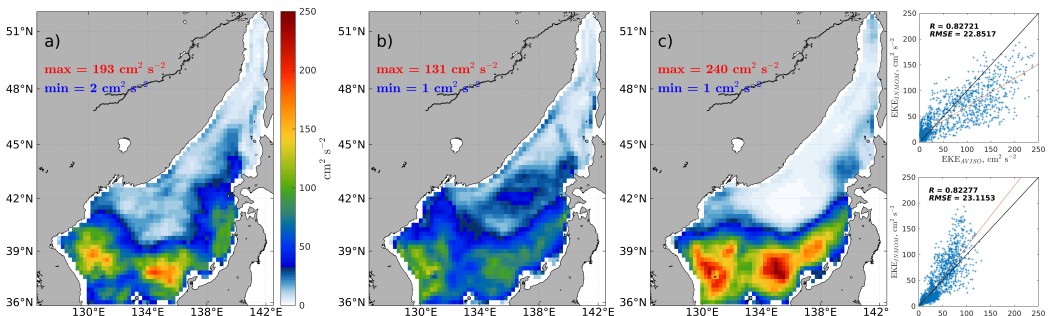

**Figure 8.** Long-term (1990–2010) annual mean eddy kinetic energy (shading, cm$^2$ s$^{-2}$) assessed with the monthly mean SSH anomalies and geostrophic relation in the Japan/East Sea from INMOM simulations (EKE$_{INMOM}$) (**a**), GOFS3.1 reanalysis (EKE$_{GOFS}$) (**b**), and AVISO dataset (EKE$_{AVISO}$) (**c**). To support the visual comparison, scatter plots with basic statistics: correlation coefficient (*R*) and root-mean-squared error (*RMSE*) are given (**d**,**e**). Monthly mean SSH from INMOM simulations and GOFS3.1 reanalysis were preliminarily interpolated on the AVISO grid.

## 4. Results

### 4.1. Simulated Mesoscale Dynamics in the Japan/East Sea

Considering the features of the mesoscale dynamics simulated with the eddy-resolving INMOM configuration, the mesoscale dynamics is mainly associated with the mesoscale eddies quantitatively characterized by the vertical component of relative vorticity (hereafter, relative vorticity). For example, we considered a snapshot of the relative vorticity field ($\zeta = \partial v/\partial x - \partial u/\partial y$) at the moment when mesoscale eddies were distinctly observed. Figure 9 shows $\zeta$ field normalized on $f$ on 1 December 2000. This field is spatially non-uniform; in the southwestern and southern parts and westward of the Tsugaru Strait, we observed anti-cyclonic/cyclonic mesoscale eddies with a negative/positive $\zeta$ and various intensity. The spatial scales of this eddy-like stricture vary from 30 km to 40 km and correspond to 2–3 $\lambda_1$, and the Rossby number ($|\zeta/f|$) has a small, but finite value.

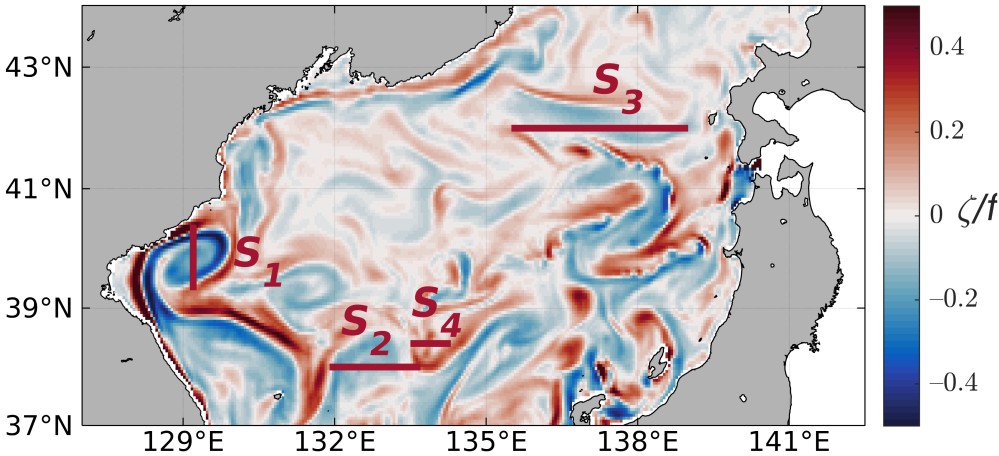

**Figure 9.** Snapshot of relative vorticity ($\zeta$) normalized by the Coriolis parameter ($f$) assessed with the simulated velocity at a depth of 15 m on 1 December 2000. Burgundy lines and S$_i$ denote transects across eddies.

Vertical sections of three anti-cyclonic eddies and one cyclonic mesoscale eddy are shown in Figure 10. Anti-cyclonic eddies have a "hot-core" with temperature anomalies relative to the ambient by 2–3 °C at 75 m. Similar salinity anomalies can be negative (sections S$_1$ and S$_2$) and positive (sections S$_3$). The cyclonic eddy has a "cold-core" with a temperature anomaly relative to the ambient by –0.5 °C at 75 m. Both anti-cyclonic and cyclonic eddies can stir isotherms and isohalines in the upper layer (see Figure 10c,d).

The typical velocity scale of these eddy-like structures varies from 5–10 cm s$^{-1}$. High-velocity anomalies indicate high-level kinetic energy associated with these mesoscale eddies. In addition, the S$_1$ and S$_3$ eddies can transport a significant volume of water from their birth locations to the northern JES.

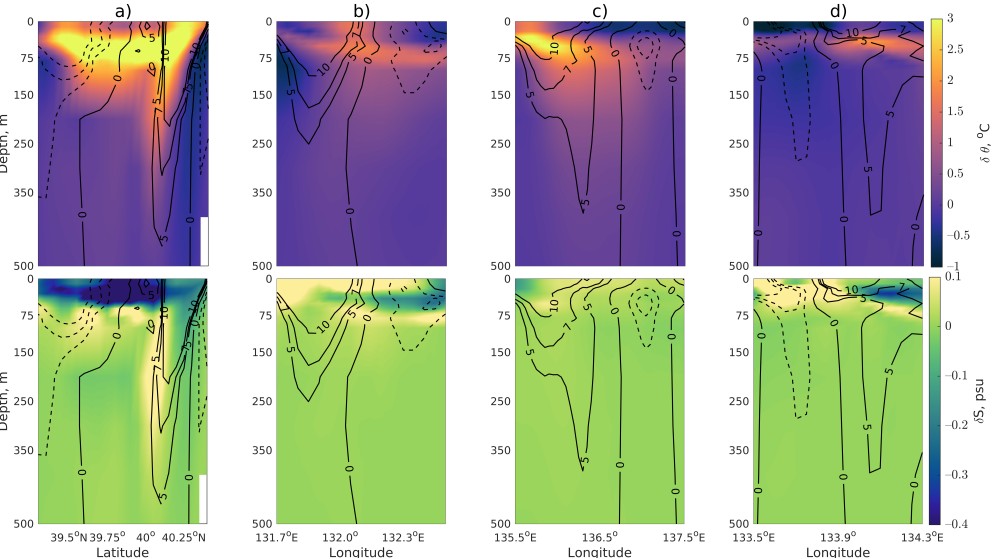

**Figure 10.** Anomalies of potential temperature (shading, $\delta\theta$, °C), salinity (shading, $\delta$S, psu), zonal ($\delta$u), and meridional ($\delta$v) velocities (black line, cm s$^{-1}$) at the vertical transections (see Figure 9) S$_1$ (**a**), S$_2$ (**b**), S$_3$ (**c**), and S$_4$ (**d**).

### 4.2. Eddy Kinetic Energy and Its Sources in the Japan/East Sea

To quantitatively characterize mesoscale eddies in the JES and analyze the mechanisms of their generation, we estimated the EKE corresponding to (3). To decompose a velocity field on mean and time-varying (or eddy) components, a monthly period was considered as a specific time of averaging. However, if the lifetimes of mesoscale eddies exceed the monthly scale, we can underestimate EKE. We estimated the velocity anomalies $\mathbf{u}'_h = (u', v')$ by subtracting 90 d running mean current velocities $(u_m, v_m)$ from the total velocity components. This time period is greater than or equal to the lifetimes of the large number of mesoscale eddies in the JES [29].

Figure 11 shows the long-term annual mean EKE profiles averaged over the northern and southern JES. In addition, we show the long-term monthly mean kinetic energy (MKE) of the mean circulation estimated with $(u_m, v_m)$ as follows

$$\text{MKE} = \frac{1}{2}\rho_0\left(\langle u_m^2\rangle + \langle v_m^2\rangle\right), \tag{4}$$

which is averaged over the year over the southern (from 33° to 41°N) and northern (from 41° to 52° N) JES.

In the southern and northern JES, region-averaged EKE reaches its maximum in the upper layers. In the southern JES, it is 1.5-times higher than the regional average EKE in the northern JES. The region-averaged EKE exceeds the region-averaged MKE in the both parts of the JES. With depth, the intensity of the mesoscale dynamics decreases faster than that of the mean circulation. These features of the region-averaged MKE and the region-averaged EKE profiles are similar to those observed in other basins of the World Ocean [11]. Nevertheless, the region-averaged EKE is comparable with the region-averaged MKE up to a depth of 300 m. This indicates that the mesoscale dynamics cover a significant upper layer. Further, we considered only the upper 300 m layer, where the region-averaged EKE has its maximum value.

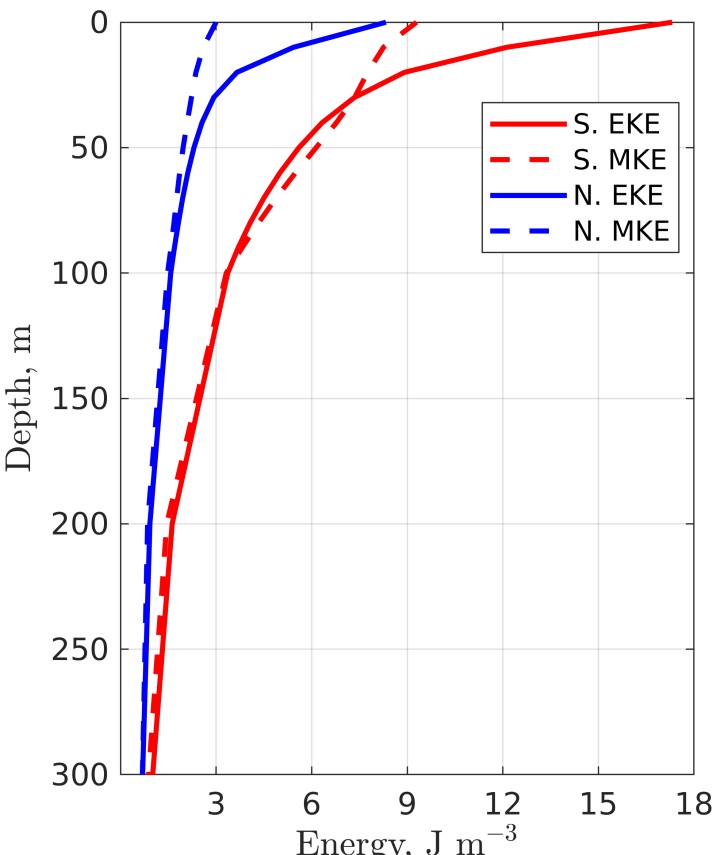

**Figure 11.** Long-term annual mean region-averaged eddy kinetic energy (solid line) and mean kinetic energy (dash line) in the southern (from 33° to 41° N) and northern (from 41° to 52° N) parts of the Japan/East Sea.

The spatial pattern of EKE integrated in the upper 300 m layer ($EKE_{300}$) and its seasonal variations are presented in Figure 12. There is a strong spatial non-uniform distribution of $EKE_{300}$ with its maximum from $2$–$4 \times 10^3$ J m$^{-2}$ in the southern JES and a minimum (less $5 \times 10^2$ J m$^{-2}$) in the northwestern JES throughout the year. The maximum of $EKE_{300}$ (more than $4 \times 10^3$ J m$^{-2}$) was observed mainly in regions covered by the three branches of the TWC.

The higher EKE was observed on the EKWC separation latitude and along the Nearshore Branch of the TWC. Furthermore, EKE reaches its higher values southward of the Yamato Rise and westward of the Tsugaru Strait. In situ observations [24,29,61] revealed numerous mesoscale eddies generated in these regions. In the northwestern JES, we observed a high EKE associated with the Primorye Current. This region of the high EKE value is associated with mesoscale eddies estimated from satellite observations. Note that in the northwestern JES, where the Primorye Current dominates, EKE shows a pronounced seasonal variability with the maximum in winter and autumn and its minimum in spring and summer. In contrast, EKE shows higher values in the southern JES during the whole year.

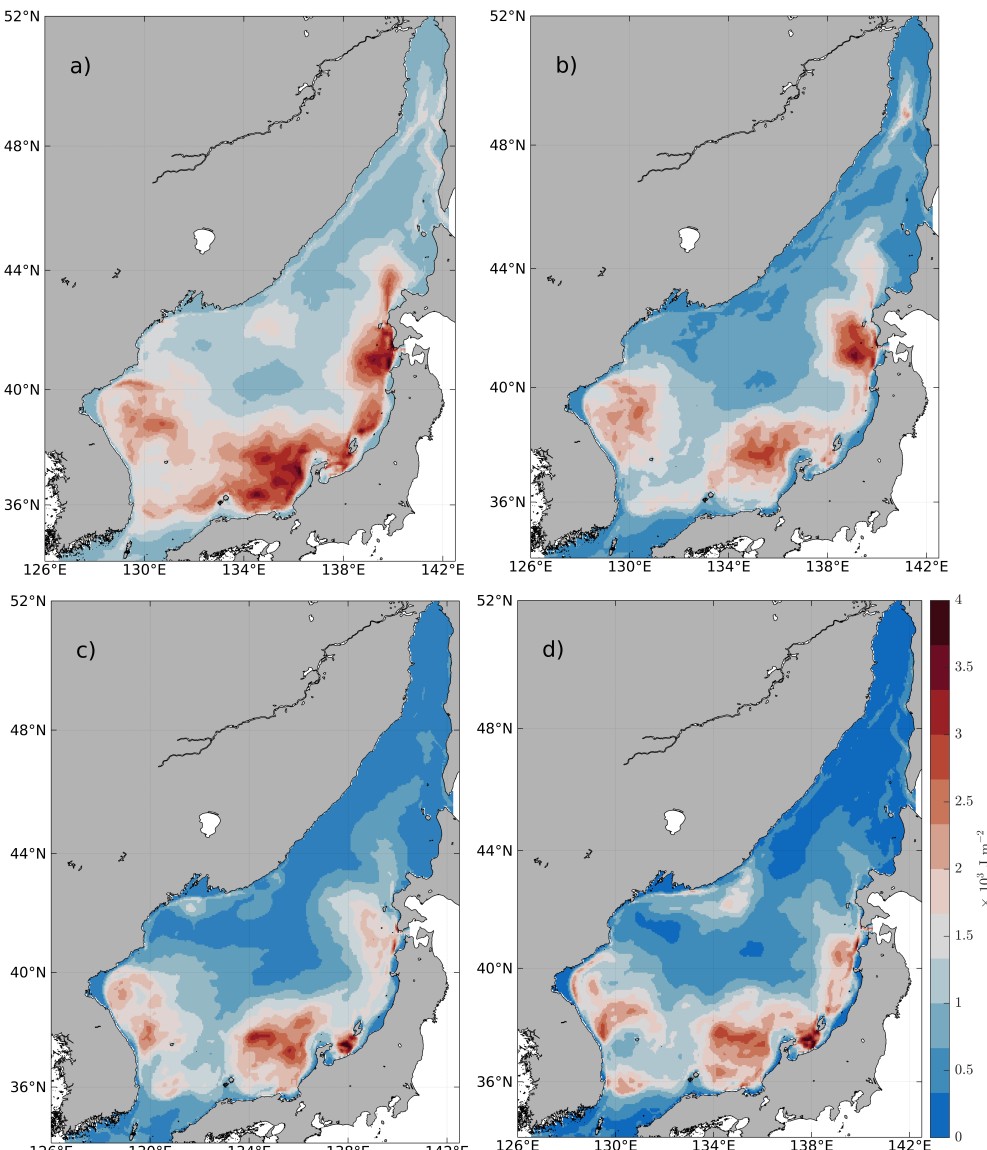

**Figure 12.** Long-term (1990–2010) seasonal-mean eddy kinetic energy (shading, $10^3$ J m$^{-2}$) integrated in the upper 300 m in winter (**a**), spring (**b**), summer (**c**), and autumn (**d**).

According to the annual mean EKE$_{300}$ distribution, we reserved four regions, where EKE$_{300}$ reaches its maximum (Figure 13). These regions are associated with three branches of the TWC, the TWC near the Tsugaru Strait, and the Primorye Current eastward of the Peter the Great Bay. This association suggests that the primary mechanism of the EKE generation could be the instability of the JES basin-scale circulation. To prove this suggestion, we considered the EKE budget in these regions.

First, we estimated the vertical profiles of the averaged EKE (aEKE) in these regions (Figure 13). For all regions, the aEKE shows its maximum up to 16 J m$^{-3}$ in the upper layer and then decreases by an order at a depth of 300 m. Note that the aEKE decreases in the Primorye Current faster than in other regions. In regions associated with the Nearshore and Offshore Branches of the TWC, the aEKE decreases slower than in other regions. In depth, aEKE profiles show various characteristics. In the intermediate and abyssal depths, the aEKE decreases in the Peter the Great Bay slower than in other regions; in the southern JES, the region-averaged EKE decreases faster than in other regions. These features of the aEKE profiles point out that in intermediate and abyssal layers, the mesoscale dynamics is associated with the instability of the basin-scale cyclonic gyre in the northern JES.

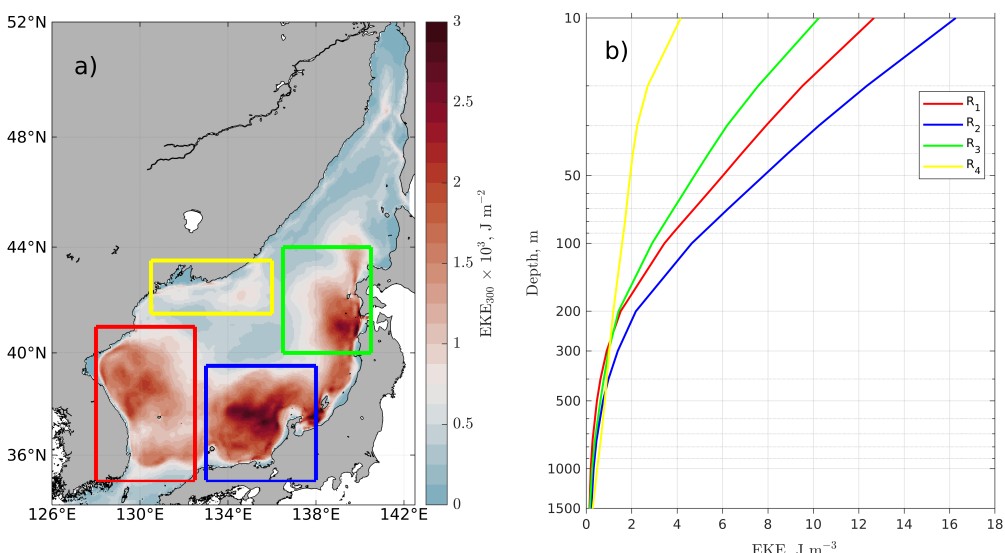

**Figure 13.** (**a**) Long-term (1990–2010) annual mean and integrated in the upper 300 m eddy kinetic energy. Bars of different colors are associated with the regions of interest: $R_1$ is denoted by red; $R_2$ is denoted by blue; $R_3$ is denoted by green; $R_4$ is denoted by yellow. (**b**) Vertical profiles of the region-averaged eddy kinetic energy.

To assess the contribution of the basin-scale circulation instability in the generation of mesoscale dynamics, we estimated the energy conversion rates in the JES. According to [11,14,19], the EKE budget can be written as follows:

$$\frac{\rho_0}{2}\left(\nabla\overline{\left(\mathbf{u}\cdot\mathbf{u}'^2_h\right)} + \frac{\overline{\partial\mathbf{u}'^2_h}}{\partial t}\right) + \nabla\cdot\overline{p'\mathbf{u}'} = -\overline{\rho'w'}g + \overline{\mathbf{u}'_h\cdot\mathbf{F}'_h} - \rho_0\overline{\mathbf{u}'_h\cdot\left(\mathbf{u}'\cdot\nabla\overline{\mathbf{u}_h}\right)}, \quad (5)$$

where $\mathbf{u} = (u, v, w)$ is three-dimensional velocity field, $w$ is the vertical velocity component, $\mathbf{F}$ includes the influence of wind and subgrid processes, $\rho'$ is the density deviation from $\rho_0$, and $p$ is pressure. In (5), the first and second terms on the left-hand side (LHS), $\frac{\rho_0}{2}\left(\nabla\overline{\left(\mathbf{u}\cdot\mathbf{u}'^2_h\right)} + \frac{\overline{\partial\mathbf{u}'^2_h}}{\partial t}\right)$ denote the change of the EKE induced by the advection of the mean current and the tendency of EKE, respectively. The third term of (5) on the LHS, $\nabla\cdot\overline{p'\mathbf{u}'}$ denotes the pressure work.

The first term of (5) on the right-hand side (RHS), $-\overline{\rho'w'}g$, denotes the rate of energy conversion from eddy potential energy (EPE) to EKE and characterizes the strength of baroclinic instability. The time-varying components of wind work and internal turbulent viscosity induced by sub-grid processes are denoted by the second term on the RHS, $\overline{\mathbf{u}'_h\cdot\mathbf{F}'_h}$. The last term of (5) on the RHS, $-\rho_0\overline{\mathbf{u}'_h\cdot\left(\mathbf{u}'\cdot\nabla\overline{\mathbf{u}_h}\right)}$, denotes the kinetic energy exchange between the mean current and eddies. We can write $-\rho_0\overline{\mathbf{u}'_h\cdot\left(\mathbf{u}'\cdot\nabla\overline{\mathbf{u}_h}\right)} = -\rho_0\left(\overline{u'^2}\frac{\partial\overline{u}}{\partial x} + \overline{v'^2}\frac{\partial\overline{v}}{\partial y} + \overline{u'v'}\left(\frac{\partial\overline{v}}{\partial x} + \frac{\partial\overline{u}}{\partial y}\right)\right) - \rho_0\left(\overline{u'w'}\frac{\partial\overline{u}}{\partial z} + \overline{v'w'}\frac{\partial\overline{v}}{\partial z}\right)$. The second term of this relation is small compared to the first term due to the small vertical velocity component relative to the horizontal ones.

According to (5), EKE sources include the baroclinic and barotropic instability of basin-scale currents and external factors associated with the wind work. To estimate the influence of these instabilities and wind work on the mesoscale dynamics, we analyzed four variables. The first variable (BC) is the energy conversion rate between the available potential energy (APE) and the eddy potential energy (EPE) [11,19,62]. We estimated BC as follows:

$$BC = -\frac{g^2}{N^2\rho_0}\left(\overline{u'\rho'}\frac{\partial\overline{\rho}}{\partial x} + \overline{v'\rho'}\frac{\partial\overline{\rho}}{\partial y}\right), \quad (6)$$

where $N$ is the buoyancy frequency. When BC is positive, APE is converted to EPE, and vice versa. The releasing EPE from APE can be converted to EKE. The strength of this energy conversion is characterized by the rate between EPE and EKE ($-\overline{\rho'w'}g$), which is considered as a second variable. Both BC and $-\overline{\rho'w'}g$ describe the energy conversion associated with baroclinic instability, but for different stages. The second variable (BT) is the energy conversion rate between MKE and EKE. BT is associated with the last term on the RHS (5) and characterizes the barotropic instability mechanism. When BT is positive, MKE is converted to EKE, and vice versa. We estimated BT as follows:

$$BT = -\rho_0\left(\overline{u'^2}\frac{\partial\overline{u}}{\partial x} + \overline{v'^2}\frac{\partial\overline{v}}{\partial y} + \overline{u'v'}\left(\frac{\partial\overline{v}}{\partial x} + \frac{\partial\overline{u}}{\partial y}\right)\right). \tag{7}$$

$-\overline{\rho'w'}g$ is considered as a third variable. The last variable of the EKE sources to be considered, $(\overline{\tau' \cdot \mathbf{u_s'}})$, is associated with the eddy wind work component and can be estimated as follows:

$$\overline{\tau' \cdot \mathbf{u_s'}}, \tag{8}$$

where $\tau' = (\tau_x', \tau_y')$ is a time-varying (turbulent) component of wind stress and $\mathbf{u_s}' = (u_s', v_s')$ is a time-varying component of sea surface velocity. We guessed that the eddy wind work can be crucial to the mesoscale dynamics' generation due to the significant wind impact on the JES circulation from autumn to winter. Note that the wind stress takes into account the difference between the 10 m wind speed and the sea surface velocity [19].

We averaged the above-mentioned variables over the reserved regions and integrated them vertically from the surface to the depth of 300 m. Figure 14 shows seasonal variations of the long-term monthly mean region-averaged and depth-integrated BC, BT, $-\overline{\rho'w'}g$, and $\overline{\tau' \cdot \mathbf{u_s}'}$. For all these regions, BC reaches its maximum from autumn to winter, and then, it decreases significantly by an order in summer (Figure 14a). Note that we observed the BC maximum in the second region, where two TWC branches prevail. The BC minimum is expressed over the fourth region associated with the Primorye Current.

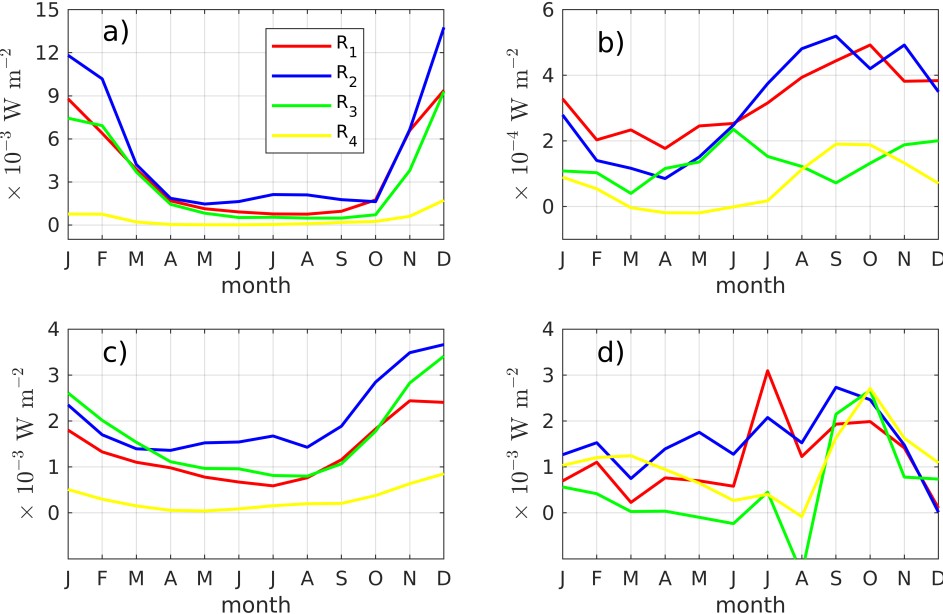

**Figure 14.** Long-term (1990–2010) monthly mean and region-averaged (see Figure 12) energy conversion rate (shading, $10^{-2}$ W m$^{-2}$) between available potential energy and eddy potential energy (BC) (**a**), mean kinetic energy and eddy kinetic energy (BT) (**b**), eddy potential energy and eddy kinetic energy ($-\overline{\rho'w'}g$) (**c**), and eddy wind work ($\overline{\tau' \cdot \mathbf{u_s'}}$) (**d**). Energy conversion rates are integrated from the sea surface to a depth of 300 m in winter.

The energy conversion rate between EPE and EKE for all the regions is shown in Figure 14c. Note that the mechanism of baroclinic instability consists of two stages. The first one is characterized by the occurrence of EPE released from APE. During the second stage, EPE can be converted to EKE. For all the regions, $-\overline{\rho'w'}g$ reaches its maximum in late summer and early autumn, that is, when EPE is intensively released from APE. $-\overline{\rho'w'}g$ reaches its maximum in the southeastern JES, where two TWC branches dominate. Note that $-\overline{\rho'w'}g$ along the TC (the third region) is higher than that in the first region, where the EKWC dominates. The minimum of $-\overline{\rho'w'}g$ was observed in the fourth region covered by the Primorye Current. Note that $-\overline{\rho'w'}g$ is three-times smaller than BC.

We show that there are pronounced seasonal variations of the energy conversion rate between APE and EPE and EPE and EKE. Baroclinic instability is associated with increasing disturbances from the thermal wind relation, which relates the vertical shear of the horizontal velocity and density gradients [63]. Changes of the vertical shear or density gradients result in the changing of the background conditions for developing baroclinic instability. Let us estimate the vertical shear (VS) of horizontal velocity variations during the year. We assessed the VS in four regions (see Figure 13) on vertical sections across currents, and then, the maximum of the VS was estimated over the whole water column excluding the mixed layer with a depth of 80 m. Figure 15 shows the seasonal variations of the mean VS for the four regions. There is a pronounced seasonal variability of the VS for all regions. The VSs reach the maxima at the end of winter and early autumn, and the VS minima are observed in March for all regions. These seasonal variations of the VS are similar to the seasonal variations of the Korean/Tsushima Strait transport of warm and salt water. Increasing this transport results in increasing the VS, density gradients, and changing of the thermal wind relation. The disturbances' relation to the thermal wind relation increases APE, which can be converted to EKE due to baroclinic instability. However the energy conversion rates (BC and $-\overline{\rho'w'}g$) are small. In autumn, when the transport across the Korean/Tsushima Strait decreases, the reducing VS results in the increasing disturbances' relation to the the thermal wind relation. Increasing disturbances reflect increasing BC and $-\overline{\rho'w'}g$. Thus, seasonal variations of the baroclinic instability intensity are associated with seasonal variations of the transport across the Korean/Tsushima Strait.

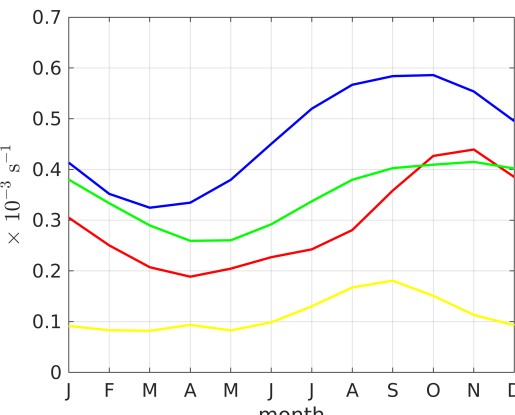

**Figure 15.** Long-term (1990–2010) monthly mean maximum of the vertical shear of the horizontal velocity on sections across currents for four regions (see Figure 9a)).

Along with the baroclinic instability, there is another EKE generation mechanism associated with the barotropic (shear) instability of basin-scale currents. The energy conversion rates between MKE and EKE are shown in Figure 14b. For three regions, BT reaches its maximum in late summer and early autumn. However, for the third region, BT reaches its maximum in summer. The BT intercomparison among the four regions shows that EKEs converted from MKE are more intensive in regions associated with the three TWC branches than in the one that is associated with the Primorye Current.

As for other EKE sources, we considered wind work, since it plays the leading role in the generation of the JES basin-scale circulation and changes its direction throughout the year. Consider the energy conversion rate between eddy wind work and EKE (Figure 14d). The intensive energy conversion was observed from late summer to early autumn when southern winds change to northern ones. For all regions, $\overline{\tau' \cdot \mathbf{u_s}'}$ reaches its maximum from September to October and then decreases. Note that a high $\overline{\tau' \cdot \mathbf{u_s}'}$ was observed in the middle of summer in the region associated with the EKWC. During the period, we observed sharply decreasing $\overline{\tau' \cdot \mathbf{u_s}'}$ in the region associated with the TC and the Primorye Current.

Based on the estimations of the energy conversion rates characterizing various sources and mechanisms of EKE generation, we established that baroclinic instability is a leading mechanism of EKE generation from winter to spring. In summer, the role of this mechanism in EKE generation weakens and the role of the barotropic instability increases. In autumn, in addition to the instability (baroclinic and barotropic) of basin-scale currents, the role of the eddy wind work affects EKE generation in the JES.

### 4.3. Horizontal Eddy Heat Transport in the Japan/East Sea

According to numerous studies [3,8,64,65], it has been established that mesoscale eddies are responsible for the transport and redistribution of the heat in the World Ocean and marginal seas. Generally, the eddy heat transport is estimated proportional to the time-averaged product of velocity and temperature deviation from their means [1]. This case does not separate the eddy heat transport on the component associated with mesoscale eddies and the one associated with mesoscale filaments and local instabilities. According to studies of the JES mesoscale dynamics [24,25,29,30,66], the mesoscale phenomena are associated mainly with mesoscale eddies and their contribution to temperature and velocity anomalies. Note that the eddy heat flux consists of rotational and divergent components [67]. The former is associated with the stirring of isotherms and does not transport heat across latitudes. In contrast, the latter is associated with the heat transporting by eddies. We assessed the meridional and zonal heat transport induced by the JES mesoscale dynamics based on the general approach.

4.3.1. Meridional Heat Transport Induced by the Mesoscale Dynamics in the Japan/East Sea.

Consider the meridional heat transport associated with the mesoscale dynamics responsible for the heat exchange between the southern and northern JES. The meridional heat transport induced by mesoscale dynamics (MEHT) is estimated as follows:

$$\text{MEHT}(x, y, t) = c_p \rho_0 \int \overline{v' \cdot \theta'} dz, \qquad (9)$$

where $c_p$ is the specific heat capacity of seawater at constant pressure equal to 4000 J (kg K)$^{-1}$.

First, we considered the seasonal changes of the zonal-integrated meridional eddy heat flux (MEHF) $\text{MEHF}(y, z, t) = c_p \rho_0 \int \overline{v' \cdot \theta'} dx$, with depth (Figure 16). During the year, the MEHF reaches its maxima in the upper 300 m layer. Beneath this layer, the MEHF decreases significantly with depth. The dominance of the positive MEHF points out that eddy heat transport is mainly polewards. We observed significant changes of the MEHF in latitude. In the southern JES, the MEHF is significantly higher than in the northern JES. The extremum of the MEHF (more than $6 \times 10^{10}$ W m$^{-1}$) is observed at the latitude, where the EKWC turns eastward to the Tsugaru Strait. Moreover, we observed seasonal variations of the extremum in the upper 100 m layer, where the MEHF reaches its maximum from autumn to winter and decreases from spring to summer.

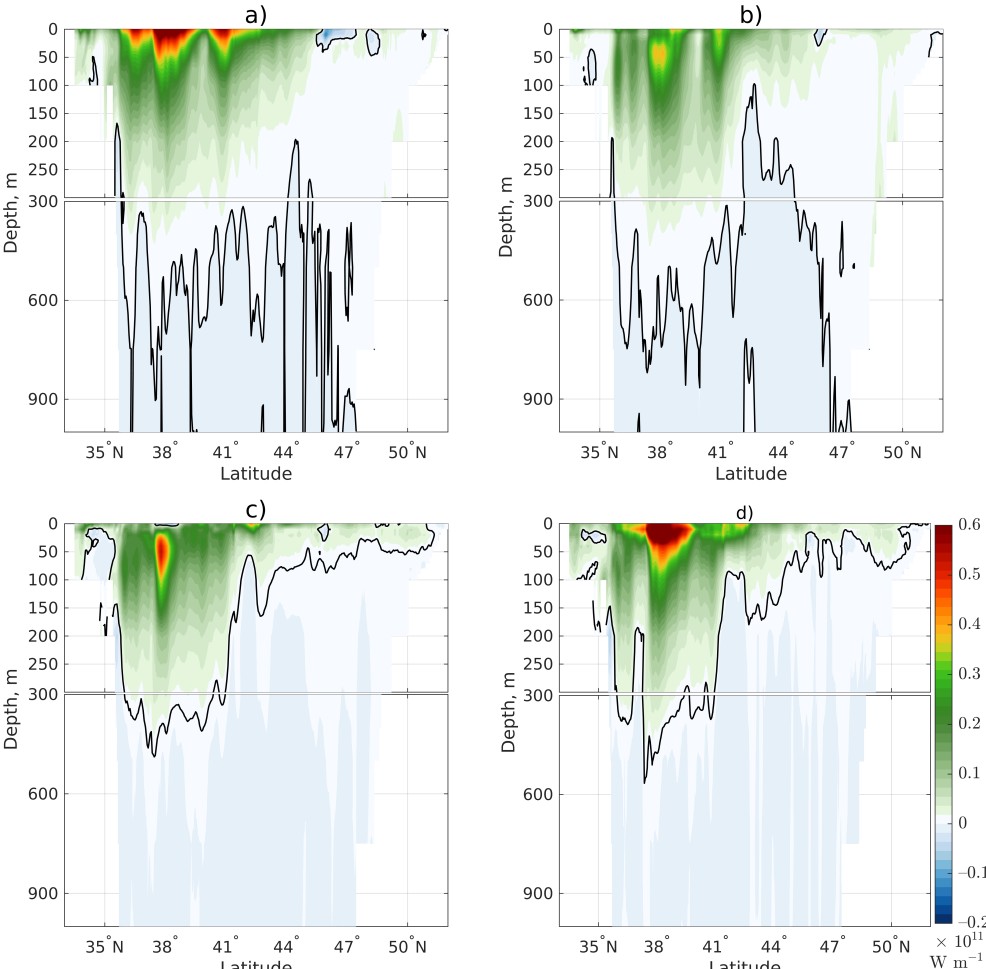

**Figure 16.** Long-term (1990–2010) seasonal mean zonal-integrated meridional heat flux driven by mesoscale dynamics in winter (**a**), spring (**b**), summer (**d**), and autumn.

To study the spatial features of the heat redistribution driven by the JES mesoscale dynamics, we considered the MEHT (see Figure 17). Since it reaches significant values in the upper 300 m layer, we considered the MEHT integrated in this layer. The MEHT spatial distribution and its seasonal variability point out the predominance of the pole-wards eddy-induced heat transport in the JES. More intensive eddy-induced heat transport takes place along the southwestern and southeastern JES boundaries, where the MEHT exceeds $0.2 \times 10^8$ W m$^{-1}$ and is associated with the TWC. This was confirmed by the revealed intensive eddy dynamics in this JES region [61]. In addition, there is an intensive eddy-induced heat transport in the southern Yamato Rise, where the MEHT exceeds $0.2 \times 10^8$ W m$^{-1}$, consistent with observations in this region [23]. The maximum MEHT is induced by eddy–heat advection in regions of the EKWC separation and over the south edge of the Yamato Rise and the region, where the Nearshore Branch merges with Offshore Branches of the TWC. We distinctly observed that the MEHT intersects the SF along the western and eastern JES. Note that the eddy-induced heat transport at 138.5° E is higher than along the western JES boundaries. A slow decrease of the MEHT northward the Tsugaru Strait is induced by the weak eddy-induced heat transport at 44° N.

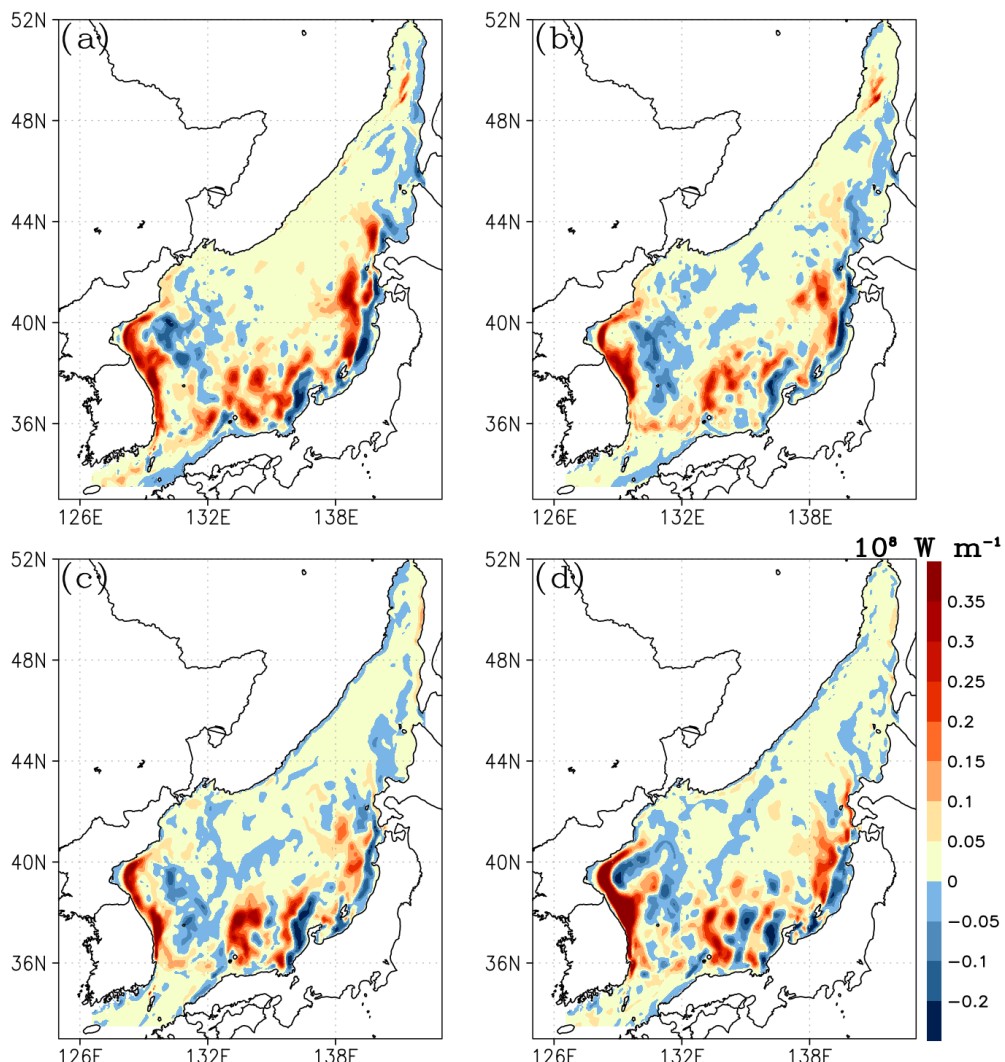

**Figure 17.** Long-term (1990–2010) seasonal mean meridional eddy heat transport in the upper 300 m in winter (**a**), spring (**b**), summer (**c**), and autumn (**d**).

At the end of this subsection, we consider the zonally integrated meridional eddy heat transport (zMEHT) in the upper 300 m layer and its seasonal variability with latitude (Figure 18). During the whole year, the zMEHT keeps positive polewards. With latitude, the zMEHT features three peaks at latitudes from 36° to 42° N, and then, it reduces to zero. At 38° N, the zMEHT reaches its maximum of 0.7–0.9 × 10$^{13}$ W, where the EKWC turns eastward. There are two peaks of the zMEHT at 37° N and 41° N equal to about 0.5 × 10$^{13}$ W. The latter peak points out eddy-induced heat transport across the SF. From autumn to winter, the zMEHT reaches its maximum at the latitude where the EKWC turns eastward, near the Tsugaru Strait and the Oki Spur. The minimum of the zMEHT is observed for all mentioned latitudes in the second half of the year. Note that northward 44°N, the MEHF decreases by more than 1 order of magnitude in relation to that in the southern JES. Thus, there is only intensive meridional eddy heat transport across the SF on its eastern and western flanks.

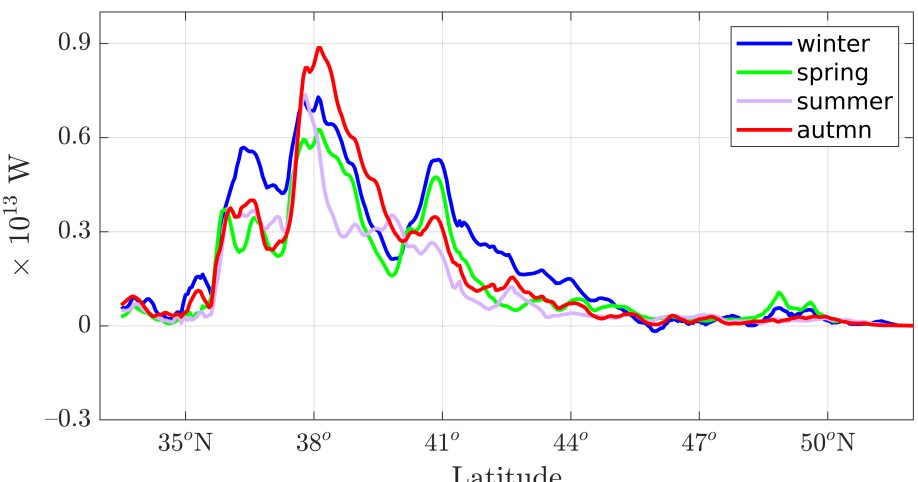

**Figure 18.** Long-term (1990–2010) seasonal mean zonally integrated meridional eddy heat transport in the upper 300 m.

### 4.3.2. Zonal Eddy Heat Transport in the Japan/East Sea.

Consider the zonal heat transport induced by the JES mesoscale dynamics (ZEHT) assessed as:

$$\text{ZEHT}(x, y, t) = c_p \rho_0 \int \overline{u' \cdot \theta'} dz. \tag{10}$$

The spatial distribution of the ZEHT in the upper 300 m and its seasonal variability is presented in Figure 19. Positive ZEHT dominates over the whole JES, and its maximum is observed on the eastern flank of the SF. In this region, the ZEHT reaches its maximum from autumn to winter and decreases in summer. During the whole year, the ZEHT is high and positive along the Offshore Branch of the TWC. Eastward of the Tsugaru Strait, the ZEHT is negative, which results from the intensive westward eddy heat transport reaching its maximum in the first half of the year and decreasing to the minimum in summer.

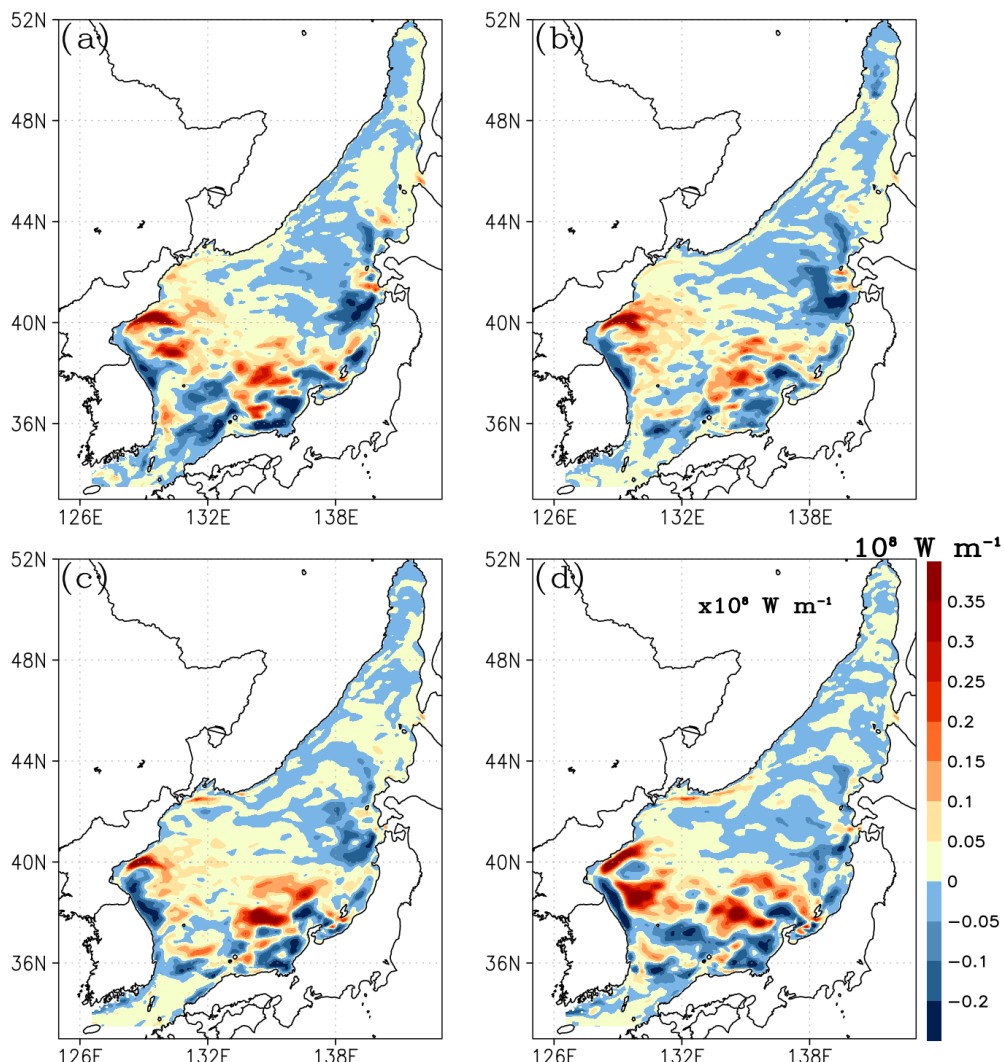

**Figure 19.** Long-term (1990–2010) seasonal mean zonal eddy heat transport in the upper 300 m in winter (**a**), spring (**b**), summer (**c**), and autumn (**d**).

## 5. Discussion

To simulate the JES mesoscale dynamics, we used atmospheric parameters from the ERA-Interim dataset of coarse spatial resolution to compute atmospheric forcing, but the ocean grid is eddy-resolving. Nevertheless, the model configuration used allowed us to reproduce major features of the JES mesoscale dynamics. The EKE spatial distribution is similar to that reproduced from floating drifter trajectories at a depth of 15 m [25,68] (see Figure 5 in [25]). The high EKE value in the southwestern and southeastern JES and westward of the Tsugaru Strait are well consistent with those assessed with the velocity dispersion from floating drifter trajectories [25]. The intensive generation of mesoscale eddies in the southeastern JES associated with the higher EKE value is consistent with the results of analyzing temperature maps [23]. During the whole year, the maximum of EKE is associated with the mesoscale eddy generation near the Oki Spur and the Noto Peninsula. Our simulations entirely confirmed the suggestion of the significant role of the hydrodynamic instability of the basin-scale circulation and particularly baroclinic instability in the JES [25]. This mechanism of the mesoscale eddy generation can be taken into account only when a high spatial resolution is used, which is twice less than $\lambda_1$ [30].

The mechanism of the baroclinic instability of the JES basin-scale circulation can also be important as other EKE sources. Neglecting this mechanism can result in overestimating other mechanisms of the mesoscale dynamics' generation. According to our simulations,

the intensive mesoscale dynamics was clearly observed along the western JES boundary. Mesoscale eddies were generated near the EKWC separation latitude during the decrease of the transport through the Korean/Tsushima Strait from late autumn and early winter, and then, these eddies propagate along the western JES boundary and transport warm and low-salinity waters polewards. These features of the mesoscale dynamics and spatial structure of currents are similar to those reproduced by ARGOS trajectories along the eastern coast of North Korea [59] and consist of a change in the velocity direction from southward to northward in this region of the JES [69]. From autumn to winter, the high EKE ($1.5$–$2.5 \times 10^3$ J m$^{-2}$) in the southwestern and northwestern JES is in a good agreement with the results of [70], where the authors revealed intensification of the mesoscale phenomenon associated with the dipole structure eastward the Peter the Great Bay. The authors declared that the leading factor of this structure is associated with the heat and freshwater transport from the southwestern JES by currents. In addition, the authors pointed out the secondary contribution of wind stress in the dipole structure generation, as indicated in [69]. Our simulations proved the secondary role of wind stress in that. We estimated the heat transport from the southwestern to northern JES and showed that the heat transport along the western JES boundary was induced by mesoscale eddies propagating polewards along this boundary. In this region, the eddy heat transport reaches $\sim 10^{13}$ W (Figure 18), which is an order of magnitude lower than estimated in [70]. Note that in the northwestern JES, the mechanism of the mesoscale dynamics' generation is the baroclinic instability of the basin-scale circulation, which was not considered in the study mentioned above. The high meridional eddy heat transport on the western flank of the SF confirms the assumption of the intensive heat exchange in this region declared in [28], where, based on the analysis of Lagrangian trajectories, the authors represented the transport of passive particle polewards by the mesoscale eddy chain.

The intensive simulated mesoscale dynamics in the northwestern JES, where the Primorye Current is located, is in a good agreement with the results of numerous studies of mesoscale dynamics in this region [66]. According to our estimations of EKE (Figure 12), the intensive mesoscale dynamics shows the pronounced seasonal variability in the northwestern JES. The maximum of EKE (up to $2.5 \times 10^3$ J m$^{-2}$) is observed in late autumn and early winter, then EKE decreases up to $\times 10^3$ J m$^{-2}$ in summer. Ladychenko et al. [66] showed that intensification of the mesoscale dynamics in autumn is associated with the dominance of northwestern winds driving a coastal upwelling, and our EKE estimations confirmed this suggestion. In autumn, an increase in the turbulent component of the wind work and an increase in shear and baroclinic instabilities lead to a more intense generation of mesoscale eddies in this region of the JES. Note that due to the small spatial size of mesoscale eddies varying from 10–20 km in the northwestern JES, the spatial resolution used is not enough to resolve the mesoscale dynamics in this region. Thus, we can underestimate the EKE in the northwestern JES.

Despite the close agreement between the simulated mesoscale dynamics and the dynamics reproduced using drifter trajectories and satellite altimetry, as well as temperature maps, the seasonal variability of the mesoscale dynamics intensity can have uncertainties associated with atmospheric parameters with a coarse spatial resolution. In particular, in winter, the high EKE can be induced by the baroclinic instability of basin-scale currents and intensive wind work and its turbulent component, which could be underestimated in our simulations. These uncertainties can be eliminated by using a high-spatial-resolution atmospheric reanalysis and regularly enlarging observational datasets spanning the study period.

The mesoscale dynamics induces meridional and zonal heat transports. Estimations of the MEHT and ZEHT show their significant values mainly in the upper 300 m layer, which confirms that the heat exchange through the Korean/Tsushima Strait plays a leading role in the generation of both the mean (basin-scale) and eddy components of the heat transport in the JES. Both components can be responsible for variations of the JES heat content. Na et al. [32] revealed the JES heat content variations based on the long-term

temperature observations from 1968 to 2007. They established that decadal variations are associated with the region westward the Tsugaru Strait. When analyzing the mechanisms responsible for this decadal variability, the authors pointed out the difficulties arising due to sporadic temperature observations in this region. Our simulations suggested that the increasing heat content in this region is induced by mesoscale dynamics inducing the intensive meridional heat transport. According to Figures 15 and 16, the meridional eddy heat transport reaches its maximum exactly in the region westward of the Tsugaru Strait. This eddy heat transport increases the TC-induced heat transport variations associated with the northwestern Pacific Ocean heat content variations [32], and jointly, these variations can result in intensive variations of the JES heat content. We are planning to comprehensively consider this mechanism of heat content variations based on retrospective simulations covering more than 30 y.

At the end of this section, we compare the meridional heat transport induced by the mean (basin-scale) circulation (MMHT) and MEHT (see Figure 20). We estimated the MMHT as:

$$\text{MMHT}(x, y, t) = c_p \rho_0 \int \langle v_m \rangle \cdot \langle \theta_m \rangle dz, \tag{11}$$

where $\langle v_m \rangle$ and $\langle \theta_m \rangle$ denote the long-term mean meridional velocity and potential temperature, respectively. We considered the vertical structure of the zonally integrated meridional heat flux (not shown) and established that it reaches its highest values in the upper 300 m layer and then decreases by more than 1 order of magnitude. Thus, we zonally integrated the MMHT from the surface to a depth of 300 m (zMMHT). It is always positive, that is, polewards, and its maximum of about 0.175 PW (1 PW = $10^{15}$ W) was observed at the latitude of the Korea/Tsushima Strait. The zMMHT maximum is associated with the inflow of subtropical waters in the JES and agrees with that assessed from in situ observations (0.17 PW) [71], as well as with the value (0.182 PW) assessed from the ocean reanalysis [72]. With latitude, the zMMHT reduces to 0.1 PW at 38° N and changes slowly from 38° to 41° N. Northward of the SF, the zMMHT reduces abruptly to 0.025 PW due to heat exchange through the Tsugaru Strait. The heat transport through the Tsugaru Strait reaches 0.085 PW, similar to the value (0.103 PW) assessed from the reanalysis [72]. Northward of 41° N, the zMMHT reduces significantly due to the heat exchange through the Soya Strait, reaching 0.02 PW. The estimated heat transport through the Soya Strait is lower than the assessed one from the reanalysis (0.034 PW) [72]. At the same time, the heat transport through the Soya Strait is similar to the one estimated from the in situ observations (about 0.025 PW) [43]. Note that the above-mentioned discrepancies between the zMMHT estimated from numerical simulations and in situ observations, as well the reanalysis in the JES Straits result from the underestimation of simulated currents (see Figures 2 and 3) and discrepancies between the simulated temperature (see Figures 4 and 5) and in situ observations, as well as reanalysis.

Quantitatively, the zMEHT is lower than the zMMHT at any latitude, and the relation of the zMEHT to the zMMHT does not exceed 10%. In particular, the zMEHT is equal to about 7.5% of the zMMHT at the EKWC separation latitude. At latitudes where two peaks of the zMEHT (Figure 18) are observed, the zMEHT is equal to 5% of the zMMHT. Although the MEHT standard deviation reaches up to 50%, the contribution of the zMEHT into the net meridional heat transport in the JES does not exceed 15% of zMMHT. Note that qualitatively, there is a significant difference between the zMMHT and the zMEHT in the southern JES. With latitude, the zMMHT reduces, and the zMEHT increases up to the latitude of the EKWC separation. Northward of this latitude, both the zMMHT and the zMEHT decrease with latitude. Since the zMEHT is very small at the latitude of the Korea/Tsushima Strait, we suggest that the heat redistribution from basin-scale currents to the mesoscale dynamics takes place mainly in the JES, and one of the primary mechanisms of this is the instability of basin-scale currents in the JES.

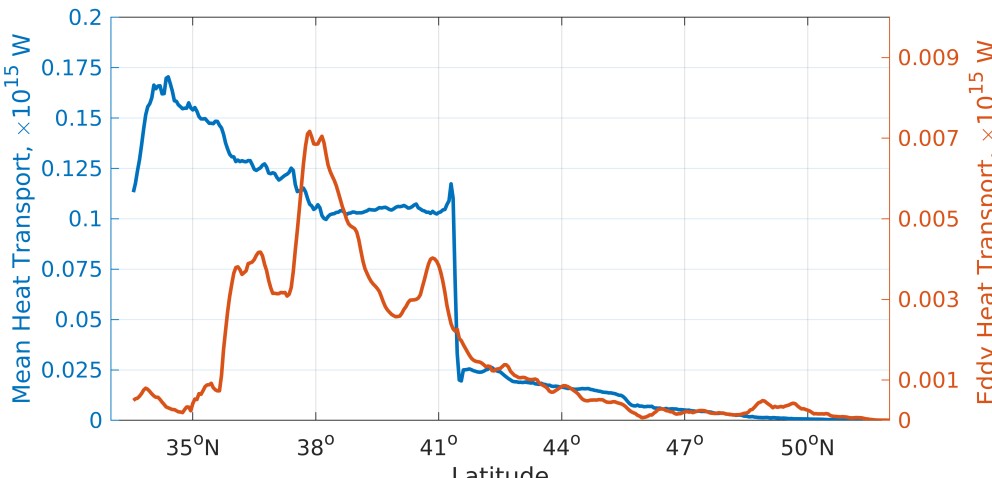

**Figure 20.** Long-term (1990–2010) annual mean meridional heat transport in the upper 300 m induced by the mean (basin-scale) circulation, denoted by blue, and the mesoscale dynamics, denoted by red. The dashed red line denotes the standard deviation of the meridional eddy heat transport.

When studying the JES climate variability, the East Sea Intermediate Water (ESIW) is considered as a marker for this variability [70]. This water is characterized by high temperature, low salinity, and a high concentration of dissolved oxygen [73]. Kim et al. [70] suggested that the generation of the ESIW is associated with mesoscale eddies. According to our investigation, the JES mesoscale dynamics is responsible for not only the intensive meridional eddy heat transport, but also eddy salt transport. In particular, Figure 21 shows snapshots of temperature and salinity anomalies at a depth of 15 m on 1 December 2000. Along the western JES boundary, in addition to the eddy heat transport, we observed the low-salinity water transport polewards induced by eddies. Moreover, we observed the salt transport induced by mesoscale eddies in the region covered by the Offshore Branch of the TWC. Warm and low-salinity waters due to the mesoscale dynamics penetrate to the SF. In winter, intensive northern and northwestern winds deliver these waters into the JES intermediate layer. Estimations of vertical heat and salt fluxes into the JES intermediate layer would be a subject of further study. Note that we need to correctly take into account the salt exchange between the East China Sea and JES when considering salt flux.

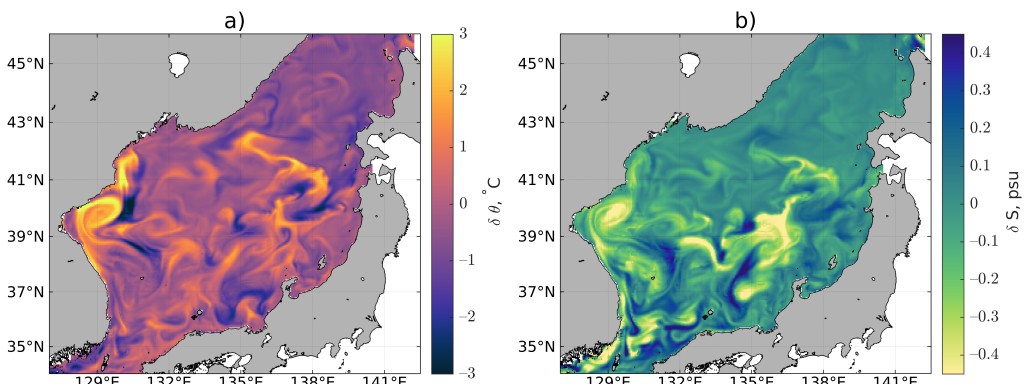

**Figure 21.** Snapshot of potential temperature ($\delta\theta$) (**a**) and salinity ($\delta S$) (**b**) anomalies at a depth of 15 m on 1 December 2000.

## 6. Conclusions

Although mesoscale eddies significantly influence the transport and redistribution of heat and salt in the Ocean, estimating this impact in oceans and marginal seas is challenging. For in situ and satellite observations, the complexity of this problem is stipulated by the

omnipresence of mesoscale eddies and their small spatial scale, in particular at high latitudes. Using eddy-resolving numerical simulations with ocean observations is a way to solve this problem.

In this study, based on retrospective eddy-resolving numerical simulations of the Japan/East Sea (JES) circulation, we investigated the mesoscale dynamics, its mechanisms, and inducing meridional and zonal heat transport from 1990 to 2010. We focused on the hydrodynamic instability of JES basin-scale currents as a primary mechanism of the mesoscale eddy generation.

Using the eddy-resolving OGCM INMOM, we reproduced the main features of the JES basin-scale and mesoscale dynamics. The specific choice of the model domain, covering the Yellow Sea, East China Sea, the southwestern part of the Sea of Okhotsk, and the northwestern part of the Pacific Ocean, allowed us to avoid the setting of boundary conditions in the JES straits and generate the subtropical water inflow through the Korean/Tsushima Strait.

We analyzed eddy kinetic energy (EKE) and established that the mesoscale dynamics was more intensive in the upper 300 m layer. Beneath this layer, EKE decreased abruptly. The spatial pattern of $EKE_{300}$ and its seasonal variations revealed the strong spatial non-uniform of $EKE_{300}$ with the maximum in the southern (southward of the Subpolar Front) JES and a minimum in the northwestern JES throughout the year. The maximum of $EKE_{300}$ was observed in regions covered by the three branches of the TWC. In the northwestern JES, EKE showed a pronounced seasonally variability with a maximum in winter and autumn and a minimum in spring and summer. We analyzed the mechanisms of the mesoscale dynamics' generation in the regions where EKE reached its maximum and confirmed that baroclinic instability was a leading mechanism of the mesoscale dynamics' generation over the JES from winter to spring. In summer, the role of this mechanism in the EKE generation weakens, and the role of the barotropic instability increases. In autumn, in addition to the instability (baroclinic and barotropic) of basin-scale currents, the role of the eddy wind work affects the EKE generation in the JES.

We estimated the meridional and zonal heat transports induced by the mesoscale dynamics in the upper 300 m layer. We found that the meridional eddy heat transport (MEHT) was mainly positive (polewards) in the JES. Low MEHT at the latitude of the Korea/Tsushima Strait indicated that the MEHT was generated mainly in the JES, and the regions of generation were associated with JES basin-scale currents. We revealed the two paths of the eddy heat transport across the Subpolar Front: along the western and eastern (along 138° E) JES boundaries. In addition, we considered the spatial distribution of the zonal eddy heat transport in the upper 300 m layer and revealed its maximum from autumn to winter and its minimum in summer. Near the Tsugaru Strait, we found an intensive westward eddy heat transport, which reached its maximum in the first half of the year and decreased to its minimum in summer.

Quantitatively, the zMEHT was very low relative to the zMMHT (no more 6–10%). At the same time, the zMEHT and zMMHT variances were of the same order. Therefore, the impact of the zMEHT on the JES heat content variability can be significant. Variations of the zMEHT and their impact on the JES heat content variability are the subject of our further studies.

**Author Contributions:** Conceptualization, N.D. and D.S.; methodology, D.S.; software, A.G., V.F., and D.S.; validation, D.S. and V.F.; writing—original draft preparation, D.S.; writing—review and editing, D.S., A.G., and V.F.; visualization, V.F. All authors have read and agreed to the published version of the manuscript.

**Funding:** Dmitry Stepanov analyzed the eddy kinetic energy budget and cross-frontal eddy-induced heat fluxes and was supported by the Russian Science Foundation (Project 19-17-00006). The numerical modeling of the eddy-resolving circulation of the Japan/East Sea was supported by the the POI FEB RAS Program "Modeling of various-scale dynamical processes in the ocean" (Project No. 121021700341-2). The estimations of the meridional eddy heat flux and its analysis were supported by the Russian Foundation for Basic Research (Project 20-05-00083). The numerical simulation outputs

were obtained using the equipment of Shared Resource Center "Far Eastern Computing Resource" IACP FEB RAS). Nikolay Diansky proposed to assess the impact of of mesoscale eddies on the heat content variability in the Japan/East Sea situated at high latitudes and was supported by the Russian Science Foundation (Project 17-17-01295).

**Institutional Review Board Statement:** Not applicable.

**Informed Consent Statement:** Not applicable.

**Data Availability Statement:** Not applicable.

**Conflicts of Interest:** The authors declare no conflict of interest. The founding sponsors had no role in the design of the study; in the collection, analyses, or interpretation of the data; in the writing of the manuscript; nor in the decision to publish the results.

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
