# Peer review of "Mesoscale Dynamics and Eddy Heat Transport in the Japan/East Sea from 1990 to 2010: A Model-Based Analysis"

_jmse, doi:10.3390/jmse10010033_

Round 1

Reviewer 1 Report

  1. English polishing is still needed. For example, the authors used both the present tense and the past tense to describe the results.

 Figure 2 caption: ‘cm2 c-2’ should be ‘cm2 s-2’;

 Line 287-289: ‘simulated (theta)’, remove the parentheses;

 Line 357: ‘three-dimensional horizontal velocity’?

  1. Figure 1a: explain the color of arrows, especially the green arrows.
  2. Model validation:

3.1 There should be a quantitative comparison on the intensity of circulation, and the KE in fig.2&3 should be changed to the magnitude of velocity. The authors attributed the stronger currents to different resolution. However, the cross-shelf temperature gradient along the Japan coast is larger in fig4a than 4b. In addition, the simulated subpolar front is much weaker than the ESA dataset in winter.

3.2 The author used the EAS dataset for T/S validation. Please add some description of this dataset and the reason that you chose it (instead of WOA2018). Since data samples are limited, higher resolution brings greater errors. I am not sure if the 1/10 grid is better for the model validation.

3.3 The validation of both temperature and salinity shows bad results north of 40N, and the simulated sea water is warmer and saltier.

The author need to check these points and reconsider the model performance.

  1. It is somewhat obscure with the understanding of regions of JES, such as the northern JES, the northwestern JES. I am not specialized in this area. For me, the northern JES may be the region which is >43N. Since the JES is not a regular shape, it is very hard to tell which part is the northern one, and a specific description or definition is essential. (Corresponding to my previous comments Q2.15&18)
  2. ‘subtracting 90-day running mean current velocities’. If the mean lifetime of mesoscale eddies in the JES is 90 days, there should be about half of eddies whose lifetime exceed 90 days. Is this reasonable to use 90 days as the boundary? Will this underestimate the EKE?
  3. Same with previous comment ‘Q2.19’. Both the BC and ‘rho*w*g’ show their maximums in winter (Figure 18a&c), and the values in winter may be twice (not sure, the authors can do the calculation) of that in spring, however, the EKE is largest in spring instead of winter in Region 2 (Figure 16b), why?

Author Response

Response to the review #1

Dear Reviewer,

Thank you for your second consideration of our manuscript " Mesoscale Dynamics and Eddy Heat Transport in the Japan/East Sea from 1990 to 2010: a Model-Based Analysis". The manuscript was revised in line with your comments. Below we list your comments together with our responses to them (in blue).

Q1.1. English polishing is still needed. For example, the authors used both the present tense and the past tense to describe the results.

A1.1. We improved our English.

Q1.2. Figure 2 caption: ‘cm2 c-2’ should be ‘cm2 s-2’;

A1.2. Corrected.

Q1.3. Line 287-289: ‘simulated (theta)’, remove the parentheses;

A1.3. Corrected.

Q1.4. Line 357: ‘three-dimensional horizontal velocity’?

A1.4. Corrected.

Q1.5. Figure 1a: explain the color of arrows, especially the green arrows.

A1.5. We added a color description in Figure 1a caption.

Model validation:

Q1.6. 3.1 There should be a quantitative comparison on the intensity of circulation, and the KE in fig.2&3 should be changed to the magnitude of velocity. The authors attributed the stronger currents to different resolution. However, the cross-shelf temperature gradient along the Japan coast is larger in fig4a than 4b. In addition, the simulated subpolar front is much weaker than the ESA dataset in winter.

A1.3. We presented quantitative comparisons of the intensity of circulation and changed the KE to the magnitude of velocity in fig.2&3.

Q1.7. 3.2 The author used the EAS dataset for T/S validation. Please add some description of this dataset and the reason that you chose it (instead of WOA2018). Since data samples are limited, higher resolution brings greater errors. I am not sure if the 1/10 grid is better for the model validation.

A1.7. We used the EAS dataset because it includes new temperature and salinity data spanning from 1804 through 2013. Our simulations covered a time period from 1979 to 2011 and we considered a period from 1990 to 2010.

Q1.8. 3.3 The validation of both temperature and salinity shows bad results north of 40N, and the simulated sea water is warmer and saltier. The author need to check these points and reconsider the model performance.

A1.8. The validation of the simulated circulation showed that it reproduced major features of the JES circulation as quantitative and qualitative. The simulated throughflow in the JES is qualitatively similar to that estimated from in-situ observations and GOFS3.1 reanalysis. Qualitative estimations of the simulated potential temperature show a closer correspondence between this temperature and one from the EAS dataset. Moreover, there is a closer correspondence between the eddy kinetic energy estimated based on the simulated geostrophic currents and one estimated from the AVISO dataset. There are discrepancies between the simulated salinity and one from the EAS dataset. However, salinity plays a minor role in our study of the mesoscale dynamics in the JES. Thus, the used model configuration is entirely applicable to the investigation of the mesoscale dynamics in the JES.   

Q1.9. It is somewhat obscure with the understanding of regions of JES, such as the northern JES, the northwestern JES. I am not specialized in this area. For me, the northern JES may be the region which is >43N. Since the JES is not a regular shape, it is very hard to tell which part is the northern one, and a specific description or definition is essential. (Corresponding to my previous comments Q2.15&18)

A1.9. We pointed out in the revised manuscript (for example, Line 204 and 2017) that the northern JES is the region located northward the Subpolar Front (>41N) and the southern JES is the region located southward the Subpolar Front (<41N). The northwestern JES is the northwestern boundary of the JES, where the Primorye Current is.

Q1.10 ‘subtracting 90-day running mean current velocities’. If the mean lifetime of mesoscale eddies in the JES is 90 days, there should be about half of eddies whose lifetime exceed 90 days. Is this reasonable to use 90 days as the boundary? Will this underestimate the EKE?

A1.10. If the monthly period is considered a specific time of averaging while the lifetime of mesoscale eddies exceeds the monthly scale, we can underestimate EKE. Thus, we use the 90-day running mean because the mean lifetime mesoscale eddies in the JES is about 90 days.

Q1.1. Same with previous comment ‘Q2.19’. Both the BC and ‘rho*w*g’ show their maximums in winter (Figure 18a&c), and the values in winter may be twice (not sure, the authors can do the calculation) of that in spring, however, the EKE is largest in spring instead of winter in Region 2 (Figure 16b), why?

A1.10. Thank you for your helpful comment. We estimated the region-averaged EKE (aEKE) and established that in the region 2, the aEKE is higher in winter then in spring. This discrepancy (between our figures) resulted from our artificial overestimate of the EKE in spring. We corrected Figure 12.

Reviewer 2 Report

Thank you for your replies and revised manuscription. I saw the changes of the figures and manuscript as well. I could see the effort of authors. However, to publish a paper in JMSE, authors need to spend time for organizing the figures. Now authors plotted 25 figures. Few of them are unnecessary for this manuscript. For example, figures no. 5, 7, 9~11 are not necessary as a validation. Especially, the salinity which plays a minor role in this study and also shows poor performance is not necessary to be shown in the current article. I ask authors to remove unnecessary figures and then it will be ready to publish in JMSE.

Author Response

Response to the review #2

Dear Reviewer,

Thank you for your second consideration of our manuscript " Mesoscale Dynamics and Eddy Heat Transport in the Japan/East Sea from 1990 to 2010: a Model-Based Analysis". The manuscript was revised in line with your comments. Below we list your comments together with our responses to them (in blue).

Q2.1. Thank you for your replies and revised manuscription. I saw the changes of the figures and manuscript as well. I could see the effort of authors. However, to publish a paper in JMSE, authors need to spend time for organizing the figures. Now authors plotted 25 figures. Few of them are unnecessary for this manuscript. For example, figures no. 5, 7, 9~11 are not necessary as a validation. Especially, the salinity which plays a minor role in this study and also shows poor performance is not necessary to be shown in the current article. I ask authors to remove unnecessary figures and then it will be ready to publish in JMSE.

A2.1. In the revised manuscript, we dropped figures no. 5, 7, 9~11.

Round 2

Reviewer 1 Report

1. The Q1.10 hasn’t been solved.   ‘subtracting 90-day running mean current velocities’. If the mean lifetime of mesoscale eddies in the JES is 90 days, there should be about half of eddies whose lifetime exceed 90 days. Is this reasonable to use 90 days as the boundary? Will this underestimate the EKE?

In other words, if there is an eddy whose lifetime is 100 days, will its kinetic energy be filtered? Maybe you can choose the period according to the power spectrum analysis of the EKE.

2. The  simulated EKE is largest in winter, however, there is obvious difference of velocity and temperature between the model results and observations (Fig 2, 4&5). The authors should add some discussion about the error of heat transport.

Author Response

Response to the review #1

Dear Reviewer,

Thank you for your third consideration of our manuscript " Mesoscale Dynamics and Eddy Heat Transport in the Japan/East Sea from 1990 to 2010: a Model-Based Analysis". The manuscript was revised in line with your comments. Below we list your comments together with our responses to them (in blue).

Q1.1. The Q1.10 hasn’t been solved.   ‘subtracting 90-day running mean current velocities’. If the mean lifetime of mesoscale eddies in the JES is 90 days, there should be about half of eddies whose lifetime exceed 90 days. Is this reasonable to use 90 days as the boundary? Will this underestimate the EKE?

In other words, if there is an eddy whose lifetime is 100 days, will its kinetic energy be filtered? Maybe you can choose the period according to the power spectrum analysis of the EKE.

A1.1. Yes, you are right. We can underestimate the EKE due to subtracting 90-day running mean current velocities. However, the number of eddies with lifetimes > 100 days is significantly less than eddies with lifetimes < 100 days (see, Lee et al. doi:10.7850/jkso.2019.24.2.267 p. 276. The number of eddies with lifetimes > 100 days equals 7 or 25% from all eddies 27). Nevertheless, your suggestion is very useful. In our future study, we will analyze the EKE, assessed from the AVISO dataset, power spectrum.

We edited our manuscript as follows (see, Line 383-387) “To quantitatively characterize mesoscale eddies in the JES and analyze mechanisms of their generation, we estimate the EKE corresponding to (3). To decompose a velocity field on mean and time-varying (or eddy) components, monthly period is considered as a specific time of averaging. However, if lifetimes of mesoscale eddies exceeds the monthly scale, we can underestimate the EKE. We estimate velocity anomalies (u',v') by subtracting 90-day running mean current velocities (um,vm) from total velocity components. This time period is greater than or equal to lifetimes of the large number of mesoscale eddies in the JES \cite{Lee2019}. ”   

Q1.2 The  simulated EKE is largest in winter, however, there is obvious difference of velocity and temperature between the model results and observations (Fig 2, 4&5). The authors should add some discussion about the error of heat transport.

A1.2. We edited our manuscript as follows (see, Line 692-696) “Note that the above-mentioned discrepancies between the zMMHT estimated from numerical simulations and in situ observations, as well reanalysis in the JES Straits result from the underestimation of simulated currents (see, Figure~2,3) and discrepancies between the simulated temperature (see, Figure~4,5) and in situ observations, as well as reanalysis.” 

This manuscript is a resubmission of an earlier submission. The following is a list of the peer review reports and author responses from that submission.

Round 1

Reviewer 1 Report

In this manuscript, the simulated mesoscale dynamics, including the eddy generation mechanism and eddy-induced heat transport, are analyzed in the Japan/East Sea (JES) from 1990 to 2010. The authors found that baroclinic instability is the leading mechanism of the eddy generation in this region through the eddy kinetic energy budget. Then, the zonal and meridional eddy-induced heat transports are estimated based on the model outputs. Furthermore, the seasonal variabilities of eddy-induced heat transport are analyzed. Although the simulated mesoscale dynamics in the JES during 1990-2010 are comprehensively analyzed, some issues still need further clarifying. Therefore, I recommend Major Revision to this paper.

Major Comments

1. In this manuscript, there are many geographical names, such as “Korea/Tsushima Strait”, “East Bay”, “Tsugaru Strait” etc., and the current names, such as “Tsushima Warm Current”, “East KoreaWarm Current”, “Primorye Current” etc.. Please consider adding necessary information in Figure 1 and more descriptions in the caption.

2. In the Introduction, many literatures have been referred to, but several have little relation with the focus of this study. For example, lines 81-88 “the submesoscale variability”, “the interannual variability in the JES surface circulation” and so on. According to the first sentence of this paragraph, references about “Numerical simulation was successfully used to reproduce the JES mesoscale dynamics” should be cited here.

3. Section 3.3 gives an estimation of the simulated EKE. In lines 276-279, it is stated that “the JES mesoscale dynamics, which is correctly reproduced”. However, the results of INMOM show a higher magnitude than that of the GOFS3.1 reanalysis but a similar magnitude with the AVISO dataset in the southern JES. Why? Can the authors discuss more about this?

4. There are two different methods used to calculate the eddy kinetic energy (equation 4 and 5), so does the kinetic energy (equation 2 and 6). The overbars also have different meanings. It may be better to put the calculation together and add more statements about using different definitions in other sections. 

5. Section 4.1 seems to be unrelated to the main focus of this study. What is its use in the whole study? Meanwhile, the three mesoscale eddies are picked randomly to me. 

6. Figure 10 gives the results of four important EKE sources. The interpretations are great, but the conclusion may be less vigorous. Line 387-390, “baroclinic instability is a leading mechanism of EKE generation in the JES throughout year”. However, the contribution of wind work seems to be similar to that of BC during August-October for R1 and R2. Please compare the results.

7. In Sections 3-4, the seasonal variability of the results occupies a great deal of the manuscript. But less is concluded in the abstract.

8. Section 5 is a little confusing and poorly organized. The aim of each paragraph is not clear to me.

Some specific comments are listed below:

  1. Line 19-20: “an individual eddy can transport warm water inside itself far from its birth location”, this refers to anticyclonic eddy? How about the cyclonic eddy?
  2. Line 39-40: Here, the “eddy-permitting” refers to what horizontal resolution? 
  3. Most of the arrows in Figures 2-3 are too small to be seen. Since the region in the east of Japan is not the focus of this study, the results here can be masked, and the reference arrow can be smaller than the 0.3m/s.
  4. Line 191-192: Please add robust evidences that “in February and July, when the transport through the Korea/Tsushima Strait reaches its minimum and maximum, respectively”. Are there any published results to support this?
  5. Line 210 and 228: the anticyclonic eddies in February and July are the same eddy?
  6. Line 328-329: According to the caption of Figure 9, it is the long-term autumn mean, not the annual mean. Is this a mistake?
  7. Line 346: What are the meanings of each term in equation 7? Would you please add more statements?
  8. Figure 12: The arrows denote for what?

Reviewer 2 Report

Based on eddy-resolving ocean model, this manuscript tried to reproduce the general circulation and mesoscale dynamics in the JES, and estimated the EKE budget and horizontal heat transport. Generally, the author concluded that the baroclinic instability is the leading mechanism of the JES mesoscale dynamics generation, and revealed some characteristics of the horizontal heat transport and their seasonality. I think that the topic of this paper is within the appropriate scope of this journal, however, there are lots of work to be done. Specific comments are described below, and I hope my comments will be of some help.

1.English polishing is indispensable.

2.There are also so many errors, such like:

 Line16:’heat absorbtion’ should be ‘heat absorption’;

 Line18:’another’ should be ‘the other’;

 Line 328: ‘annual mean’, but in figure 9, it is ‘autumn mean’;

 Line 346: rho should be rho’; ‘we analyze three variable’, should be ‘four variable’’?

 Line 550&564: ‘ZMMH’ should be ‘ZMMHT’.

In addition, ‘the subsurface *meter’ should be ‘the upper *meter’; What is ‘BT’, ‘BC’, ‘Us in Eq.11’?

3.The abbreviations of some journals’ name are incorrect. In addition, is it necessary to refer so many papers? Please refine the references.

4.The authors should show the schematic of the general (or seasonal) circulation in JES in one or two figures, and label the circulation names, strait names, countries, as well as the subpolar front in these figures.

Figure 4: The order of subplot is different with figure2&3, and a consistent order in different figures should be better.

Figure 12: what is the arrow represent?

5.Model validation: There should be more quantitative comparison on the intensity of circulation and EKE with figure 2-4. Considering the EKE budget, heat transport, and their seasonality, the simulated temperature and salinity also should be validated in the upper 300-m (SST, key sections, seasonality). How about the seasonality of the simulated througflow in the three sections?

There should be some explanation of the parameter configuration in Table 1. How does the author get these numbers? As I see, the simulated current looks stronger than altimetry data (figure 2), maybe there are still some improvement of the model configuration.

7.Line 265-266: The author said both simulated SSH and GOFS show high EKE in the northern JES. This is obscure. The high EKE shows in the southern JES(line 260-261).What is the meaning about the ‘high value’, and where is the ‘northern JES’?

8.The author showed three anticyclonic eddies in section 4.1, and it’s better to introduce both anticyclonic and cyclonic eddy.

9.Line 304: ‘after running averaging with the window of 90days’, why 90days?

  1. Line 324-326: maximum in autumn?

11.The EKE in Region2 shows its maximum in spring (figure 8b), however, the BC in Region 2 shows its maximum in winter (Figure 10a), why? Since the BC is the leading mechanism, the EKE and BC should be basically consistent, right?

12.I suggest further discussion about the vertical shear of background velocity in different sections, which helps us to understand the EKE budget and the baroclinic instability. The paper listed below is suggested.

Qiu, B., 1999: Seasonal eddy field modulation of the North Pacific Subtropical Countercurrent: TOPEX/POSEIDON observations and theory. J. Phys. Oceanogr., 29, 2471–2486.

13.Detailed numerical model validation was needed (mentioned above), which can also improve the accuracy and reliability of the quantitative estimation of eddy induced heat transport.

Reviewer 3 Report

This manuscript presents an analysis based on model simulation of mesoscale dynamics and eddy heat transport in the Japan/East sea from 1990 to 2010.

This paper is well written and the analysis seems reasonable dealing with scientific issue. This paper can be published after few modifications of the manuscript which suggested as below.

Authors are discussing about the eddy kinetic energy and meridional heat transport in the Japan/East Sea. For the eddy kinetic energy and heat transport, the vertical structure of the temperature and the ocean current play a crucial role. Therefore, authors need to validate  more about the ocean current and vertical structures of the ocean comparing with an observation quantitatively(Only long term averaged transport/current validation is not enough). Also the model used in this study shows strong overshoot of warm current along the eastern side of Korean peninsula which is not shown in either observation or GOFS3.1. It seems like authors need to deal with this issue first and analyze the result.

Reviewer 4 Report

Review of "Mesoscale Dynamics and Eddy Heat Transport in the Japan/East Sea from 1990 to 2010: a Model-Based Analysis" (Stepanov et al., Manuscript ID: jmse-1462008)

The objective of the manuscript is aimed at studying driving mechanisms of mesoscale processes and related heat transport in the Japan/East Sea from 1990 to 2010 using an eddy-resolving numerical model. If I do not misunderstand, the authors declare that previous studies did not find leading mechanisms of the mesoscale dynamics and heat transport induced by mesoscale eddies. Based on results from numerical simulations, they conclude that (1) mesoscale eddies can deepen isotherms/isohalines at an order of magnitude ~100 m and transport warm and low salinity waters along the western and eastern boundaries, (2) baroclinic instability is the leading mechanism of the mesoscale dynamics, (3) meridional heat transport is mainly poleward, (4) two eddy heat transport paths across the Subpolar Front: one along the western and the other along eastern (138°E) boundaries, and (5) an intensive westward eddy heat transport is found reaching its maximum in the first half of the year and decreasing to minimum by summer. First of all, after reading the manuscript, I am not sure if the other reviewers have the same problem as mine: the usage of English is awkward and thus somewhat disturbed the review of this manuscript. Besides the language problem, the authors should quantitatively compare their model results to those from the other sets of products instead of qualitative descriptions for the associated comparisons. The methodology and analyses are somewhat not new. I would suggest that the authors conduct a thorough revision, including the usage and style of English, and strengthen what the novelty of this study is before the manuscript can be further considered to publish in JMSE. My comments are summarized as follows.

L. 1-3: The first sentence in Abstract is awkward. Consider if "Driving mechanisms of mesoscale processes and associated heat transport in the Japan/East Sea (JES) during 1990 through 2010 are examined using eddy-resolving ocean model simulations." is better.

L. 5: I am not sure what "at a hundred meters" means here.

L. 19-20: Whether or not a mesoscale eddy can transport watermass is also discussed in Early et al. (2011, JPO, doi:10.1175/2011JPO4601.1).

L. 39-40: Since computer cluster developed quickly over past two decades, I do not think downscale computing of nested grid models is expensive nowadays.

L. 136-137: It is not clear to me what "smoothed by 9-point filter three times" is.

L. 199-212: The expression for the comparisons of geostrophic flows and kinetic energy obtained from this study, AVISO satellite altimetry, and GOFS3.1 reanalysis is qualitative. "Similar" is actually not a good term to convince readers how good the model reproduced ocean circulation is. The authors should provide quantitative comparisons, such as goodness of fit (to which one of the flow field products) or discrepancy between the three products, to demonstrate the reliability of their numerical simulations.

Figure 2 between L. 212 and 213: Confirm if the unit "cm2 c2" is correct. The expression of "log10(cm2 c2)" is also problematic. Figure 3 has the same problems. The authors should clarify these.

L. 276-277: I am not comfortable with the statement that the authors declare the mesoscale dynamics is correctly reproduced. The authors should provide solid discussions to convince this statement to readers.

Figure 8: The shading unit (103 J m2) in the figure caption is not consistent with that (104  J m2) in the figure.

P. 15, first line: Change "Is is associated..." to "It is associated...".

Figure 10: Some of the descriptions in the caption do not match with the figure.

L. 604-619: I suggest the authors to delete this paragraph because the model produced fields have been compared with those from AVISO and GOFS3.1 mostly in Section 3.